# SAMPLE-EFFICIENT AND SCALABLE EXPLORATION IN CONTINUOUS-TIME RL

**Klemens Iten, Lenart Treven, Bhavya Sukhija, Florian Dörfler, Andreas Krause**
ETH Zürich, Zürich, Switzerland
`{kiten, trevenl, sukhijab, dorfler, krausea}@ethz.ch`

## ABSTRACT

Reinforcement learning algorithms are typically designed for discrete-time dynamics, even though the underlying real-world control systems are often continuous in time. In this paper, we study the problem of continuous-time reinforcement learning, where the unknown system dynamics are represented using nonlinear ordinary differential equations (ODEs). We leverage probabilistic models, such as Gaussian processes and Bayesian neural networks, to learn an uncertainty-aware model of the underlying ODE. Our algorithm, COMBRL, greedily maximizes a weighted sum of the extrinsic reward and model epistemic uncertainty. This yields a scalable and sample-efficient approach to continuous-time model-based RL. We show that COMBRL achieves sublinear regret in the reward-driven setting, and in the unsupervised RL setting (i.e., without extrinsic rewards), we provide a sample complexity bound. In our experiments, we evaluate COMBRL in both standard and unsupervised RL settings and demonstrate that it scales better, is more sample-efficient than prior methods, and outperforms baselines across several deep RL tasks.

## 1 INTRODUCTION

Reinforcement learning (RL) has proven to be a flexible paradigm for learning control policies through interaction, with success stories across robotics (Levine et al., 2016; Hwangbo et al., 2019; Spiridonov et al., 2024), games (Schrittwieser et al., 2020; Hafner et al., 2025), and applications in medicine and energy (Yu et al., 2021; Degrave et al., 2022). In most RL algorithms, time is discretized: agents select actions at fixed intervals and the system evolves in discrete steps. However, many real-world control systems, from physical robots to biological processes, are naturally modeled by continuous-time dynamics governed by ordinary differential equations (ODEs). Discretization can obscure key temporal behaviours and limit control flexibility, whereas continuous-time models align better with real-world sensing and actuation.

In this work, we study *continuous-time model-based reinforcement learning*, where the goal is to interact with an unknown dynamical system and use the data collected on the system to learn the underlying ODE. We address two settings: (*i*) reward-driven RL, where the goal is to solve a specific task; and (*ii*) unsupervised RL, where the objective is to learn the system dynamics globally. The latter requires accurate global modeling, while the former demands task-relevant accuracy.

We propose **COMBRL**, a continuous-time model-based RL algorithm that uses probabilistic models, e.g., Gaussian processes and Bayesian neural networks, to capture the epistemic uncertainty of the learned model. Policies are selected by maximizing a weighted sum of extrinsic reward and epistemic uncertainty, following the optimism-in-the-face-of-uncertainty principle (Curi et al., 2020; Kakade et al., 2020; Treven et al., 2023; Sukhija et al., 2025b). In the unsupervised setting often encountered in system identification (Åström and Eykhoff, 1971) and in unsupervised exploration (Aubret et al., 2019; Sekar et al., 2020; Sukhija et al., 2023), **COMBRL** reduces to active learning (Taylor et al., 2021), specifically uncertainty sampling (Lewis and Catlett, 1994) and guides the agent toward regions where the model is most uncertain.

We show that **COMBRL** achieves sublinear regret in the reward-driven case and provide sample complexity bounds in the unsupervised setting. Empirically, we evaluate **COMBRL** on several deep continuous-time RL tasks, demonstrating improved performance and sample efficiency compared to baselines.

Unlike discrete-time RL, where interaction occurs on a fixed grid and uncertainty is evaluated only at visited steps, continuous-time reinforcement learning requires the agent to additionally decide when to observe and control the system. As a result, uncertainty accumulates along entire trajectories, and regret depends not only on the policy but also on the measurement schedule. Prior continuous-time approaches typically enforce optimism by optimizing jointly over policies and plausible dynamics, which is computationally demanding. In contrast, **COMBRL** realizes optimism through an intrinsic reward and directly trades off reward and epistemic uncertainty, yielding a scalable exploration mechanism whose guarantees explicitly depend on the measurement strategy.

### Contributions

- We propose **COMBRL**, a continuous-time optimistic model-based RL algorithm that balances task performance and model exploration. Unlike prior continuous-time methods, which are either purely exploitative or rely on costly co-optimization, **COMBRL** uses a single scalar to balance reward and epistemic uncertainty, supporting both reward-driven and unsupervised learning.
- We provide theoretical guarantees, showing sublinear regret in the reward-driven case and offer sample complexity bounds in the unsupervised setting, with explicit dependence on the measurement selection strategy.
- We demonstrate strong empirical results across continuous-time deep RL benchmarks, showing that **COMBRL** scales better than prior methods and generalizes to unseen downstream tasks.

## 2 PROBLEM SETTING

Consider an unknown continuous-time dynamical system $\boldsymbol{f}^*(\boldsymbol{x}(t), \boldsymbol{u}(t))$ with initial state $\boldsymbol{x}(0) = \boldsymbol{x}_0 \in \mathcal{X} \subset \mathbb{R}^{d_x}$ and control input $\boldsymbol{u} : [0, \infty) \to \mathcal{U} \subset \mathbb{R}^{d_u}$. The state at time $t$ is obtained by integrating the deterministic dynamics:

$$\boldsymbol{x}(t) = \boldsymbol{x}_0 + \int_0^t \boldsymbol{f}^*\big(\boldsymbol{x}(s), \boldsymbol{u}(s)\big)\, ds.$$

Furthermore, consider a policy $\boldsymbol{\pi} : \mathcal{X} \to \mathcal{U}$ so that the control input follows said policy, i.e., $\boldsymbol{u}(t) = \boldsymbol{\pi}(\boldsymbol{x}(t))$. Typically in optimal control (OC, cf. Luenberger (1979)), we consider the associated finite-time OC problem over the policy space $\Pi$ to solve for the optimal policy such that an objective function $J(\boldsymbol{\pi}, \boldsymbol{f}^*)$ is maximized:

$$
\begin{aligned}
\boldsymbol{\pi}^* &\stackrel{\text{def}}{=} \arg\max_{\boldsymbol{\pi} \in \Pi} J(\boldsymbol{\pi}, \boldsymbol{f}^*) = \arg\max_{\boldsymbol{\pi} \in \Pi} \int_0^T r\big(\boldsymbol{x}(s), \boldsymbol{\pi}(\boldsymbol{x}(s))\big)\, ds \\
\text{s.t.} \quad &\dot{\boldsymbol{x}}(t) = \boldsymbol{f}^*\big(\boldsymbol{x}(t), \boldsymbol{\pi}(\boldsymbol{x}(t))\big), \quad \boldsymbol{x}(0) = \boldsymbol{x}_0, \quad \big(\boldsymbol{x}(t), \boldsymbol{u}(t)\big) \in \mathcal{X} \times \mathcal{U}, \quad t \in [0, T].
\end{aligned}
\tag{1}
$$

To gather information about the unknown dynamics $\boldsymbol{f}^*(\boldsymbol{x}(t), \boldsymbol{u}(t))$, we collect data over episodes $n \in 1, \dots, N$. In each episode, we select a policy $\boldsymbol{\pi}_n \in \Pi$ by optimizing an objective (e.g., task reward or exploration) and deploy it for the horizon $T$.

Optimizing solely for the reward function $r(\boldsymbol{x}(t), \boldsymbol{u}(t))$ during learning introduces a directional bias, since $\boldsymbol{f}^*$ is mostly explored in high-reward regions of the state-action space, resulting in limited exploration and poor generalization of the learned dynamics. However, in many practical scenarios, such as unsupervised RL or system identification, the objective is not to optimize a performance criterion, but to learn the underlying nonlinear ODE $\boldsymbol{f}^*$ as fast and accurately as possible.

In practice, during an episode $n$, we query $\boldsymbol{f}^*(\boldsymbol{x}(t), \boldsymbol{u}(t))$ by taking a sequence of measurements at $m_n$ selected time points specified by a *measurement selection strategy (MSS)* $S = (S_n)_{n \geq 1}$:

**Definition 1** (Measurement selection strategy, Treven et al. (2023)). *A measurement selection strategy $S$ is a sequence of sets $(S_n)_{n \geq 1}$, such that $S_n$ contains $m_n$ points at which we take measurements, i.e., $S_n \subset [0, T], |S_n| = m_n$.*[1]

---

[1] Here, the set $S_n$ may depend on observations prior to episode $n$ or is even constructed while we execute the trajectory. For ease of notation, we do not make this dependence explicit.

This gives us a dataset of measurements $\mathcal{D}_n \sim (\boldsymbol{\pi}_n, S_n)$ as follows:

$$\mathcal{D}_n \stackrel{\text{def}}{=} \{(\boldsymbol{z}_n(t_{n,i}), \dot{\boldsymbol{y}}_n(t_{n,i})) \mid t_{n,i} \in S_n, i \in \{1 \ldots, m_n\}\} \qquad \text{where}$$

$$\boldsymbol{z}_n(t_{n,i}) \stackrel{\text{def}}{=} \big(\boldsymbol{x}_n(t_{n,i}), \boldsymbol{\pi}_n(\boldsymbol{x}_n(t_{n,i}))\big) \in \mathcal{Z} = \mathcal{X} \times \mathcal{U}, \quad \dot{\boldsymbol{y}}_n(t_{n,i}) \stackrel{\text{def}}{=} \dot{\boldsymbol{x}}_n(t_{n,i}) + \boldsymbol{\epsilon}_{n,i}.$$

Since direct measurement of $\dot{\boldsymbol{x}}$ may not be feasible, it is typically estimated via finite differences or filtering, and is subject to measurement noise $\boldsymbol{\epsilon}_{n,i}$. Thus, at episode $n$, the collected dataset up to the current episode $\mathcal{D}_{1:n-1} := \bigcup_{i=1}^{n-1} \mathcal{D}_i$ informs the policy $\boldsymbol{\pi}_n$ deployed for horizon $T$. Note that, because we defined a continuous-time setting, we are not restricted to a fixed sampling rate and **COMBRL** allows flexible, event-driven sampling and control.

## 2.1 RELATED WORK

We briefly mention the most relevant related works below. An extended discussion of related works is provided in Appendix A.

Yildiz et al. (2021) introduce a continuous-time model-based RL approach that greedily maximizes the reward integral. Treven et al. (2023) show that greedy exploration performs suboptimally and propose leveraging optimistic dynamics to encourage exploration in uncertain regions. The resulting algorithm, OCORL, enjoys convergence guarantees under common continuity assumptions on the underlying ODE. However, the method does not scale to higher dimensional systems since it relies on a challenging optimization over the set of plausible dynamics to enforce optimism.

Moreover, both aforementioned works rely on an external reward signal. When no extrinsic reward is available, or when accurate dynamics modeling is prioritized over extrinsic performance, these methods cannot be applied. On the other hand, intrinsic motivation techniques have long been used for this purpose (Salge et al., 2014; Bellemare et al., 2016; Pathak et al., 2017; Sekar et al., 2020). In particular, Sukhija et al. (2023) show convergence of intrinsic exploration methods in discrete time. However, their continuous-time counterpart is much less understood.

To this end, we present **COMBRL**, a scalable and efficient continuous-time model-based RL algorithm with a flexible optimism-driven objective that enables a natural transition between reward-driven (extrinsic) and uncertainty-driven (intrinsic) exploration. Unlike prior continuous-time methods, which are only designed for extrinsic exploration, **COMBRL** supports both extrinsic and intrinsic settings. In addition, unlike Treven et al. (2023), **COMBRL** also scales to higher dimensional settings since it does not require optimizing over the set of plausible dynamics. Our work is closely related to Sukhija et al. (2025b), who study the problem in the discrete-time setting. However, our focus on continuous-time systems requires different theoretical analysis and experimental design, as we discuss in Sections 3 and 4.

## 2.2 PERFORMANCE MEASURE

For the supervised RL setting, a natural performance measure is given by the *cumulative regret* that sums the gaps between the performance of the policy $\boldsymbol{\pi}_n$ at episode $n$ and the optimal policy $\boldsymbol{\pi}^*$ over all the episodes:

$$R_N \stackrel{\text{def}}{=} \sum_{n=1}^{N} r_n \stackrel{\text{def}}{=} \sum_{n=1}^{N} J(\boldsymbol{\pi}^*, \boldsymbol{f}^*) - J(\boldsymbol{\pi}_n, \boldsymbol{f}^*) \tag{2}$$

If the cumulative regret $R_N$ is sublinear in $N$, then the average reward of the policy converges to the optimal reward, and by extension to the optimal policy $\boldsymbol{\pi}^*$.

## 2.3 ASSUMPTIONS

In the following, we make some common assumptions that allow us to theoretically analyse the regret $R_N$ and prove a regret bound. We first make an assumption on the continuity of the underlying system and the observation noise.

**Assumption 1** (Lipschitz continuity). *The dynamics model $\boldsymbol{f}^*$, reward $r$, and all policies $\boldsymbol{\pi} \in \Pi$ are $L_f$, $L_r$ and $L_{\boldsymbol{\pi}}$ Lipschitz-continuous, respectively.*

**Assumption 2** (Sub-Gaussian noise). *We assume that the measurement noise $\boldsymbol{\epsilon}_{n,i}$ is i.i.d. $\sigma$-sub Gaussian.*

The Lipschitz assumption is commonly made for analysing nonlinear systems (Khalil, 2014) and is satisfied for many real-world applications. Furthermore, assuming $\sigma$-sub Gaussian noise (Rigollet and Hütter, 2023) is also fairly general and is common in both RL and Bayesian optimization literature (Srinivas et al., 2012; Chowdhury and Gopalan, 2017).

In **COMBRL**, we learn an uncertainty-aware model of the underlying dynamics. Therefore, we obtain a mean estimate $\boldsymbol{\mu}_n(\boldsymbol{z})$ and quantify our epistemic uncertainty $\boldsymbol{\sigma}_n(\boldsymbol{z})$ about the function $\boldsymbol{f}^*$:

**Definition 2** (Well-calibrated statistical model of $\boldsymbol{f}^*$, Rothfuss et al. (2023)). *Let $\mathcal{Z} \stackrel{def}{=} \mathcal{X} \times \mathcal{U}$. An all-time well-calibrated statistical model of the function $\boldsymbol{f}^*$ is a sequence $\{\mathcal{M}_n(\delta)\}_{n \geq 0}$, where*

$$\mathcal{M}_n(\delta) \stackrel{def}{=} \left\{ \boldsymbol{f} : \mathcal{Z} \to \mathbb{R}^{d_x} \mid \forall \boldsymbol{z} \in \mathcal{Z}, \forall j \in \{1, \ldots, d_x\} : |\mu_{n,j}(\boldsymbol{z}) - f_j(\boldsymbol{z})| \leq \beta_n(\delta)\sigma_{n,j}(\boldsymbol{z}) \right\},$$

*if, with probability at least $1 - \delta$, we have $\boldsymbol{f}^* \in \bigcap_{n \geq 0} \mathcal{M}_n(\delta)$. Here, $\mu_{n,j}$ and $\sigma_{n,j}$ denote the $j$-th element in the vector-valued mean and standard deviation functions $\boldsymbol{\mu}_n$ and $\boldsymbol{\sigma}_n$ respectively, and $\beta_n(\delta) \in \mathbb{R}_{\geq 0}$ is a scalar function that depends on the confidence level $\delta \in (0, 1]$ and which is monotonically increasing in $n$.*

**Assumption 3** (Well-calibration). *We assume that our learned model is an all-time well-calibrated statistical model of $\boldsymbol{f}^*$. We further assume that the standard deviation functions $(\boldsymbol{\sigma}_n(\cdot))_{n \geq 0}$ are $L_{\boldsymbol{\sigma}}$-Lipschitz continuous.*

Intuitively, Assumption 3 states that we are, with high probability, able to capture the true dynamics $\boldsymbol{f}^*$ within a confidence set spanned by our predicted mean $\boldsymbol{\mu}_n$ and epistemic uncertainty $\boldsymbol{\sigma}_n$ and in turn, **COMBRL** learns a probabilistic model $\boldsymbol{f}_n$ that provides both mean predictions $\boldsymbol{\mu}_n(\boldsymbol{z})$ and uncertainty estimates $\boldsymbol{\sigma}_n(\boldsymbol{z})$. For Gaussian process (GP) models, the assumption is satisfied (Rothfuss et al., 2023, Lemma 3.6), and for more general classes of models such as Bayesian neural networks (BNNs), re-calibration techniques (Kuleshov et al., 2018) can be used. Thus, **COMBRL** is model-agnostic: the statistical model can be instantiated using GPs (Rasmussen and Williams, 2005; Deisenroth et al., 2015), BNNs (MacKay, 1992), ensembles (Lakshminarayanan et al., 2017), or other estimators that capture epistemic uncertainty.

Lastly, we make an assumption on the regularity of the dynamics by placing them in a reproducing kernel Hilbert space (RKHS), which enforces smoothness and boundedness:

**Assumption 4** (RKHS Prior on Dynamics). *We assume that the functions $f_j^*$, $j \in \{1, \ldots, d_{\boldsymbol{x}}\}$ lie in a RKHS with kernel $k$ and have a bounded norm $B$, that is*

$$\boldsymbol{f}^* \in \mathcal{H}_{k,B}^{d_{\boldsymbol{x}}}, \quad with \quad \mathcal{H}_{k,B}^{d_{\boldsymbol{x}}} = \{\boldsymbol{f} \mid \|f_j\|_k \leq B, j = 1, \ldots, d_{\boldsymbol{x}}\}.$$

*Moreover, we assume that $k(\boldsymbol{z}, \boldsymbol{z}) \leq \sigma_{\max}$ for all $\boldsymbol{x} \in \mathcal{X}$.*

## 3 COMBRL: CONTINUOUS-TIME OPTIMISTIC MODEL-BASED RL

We now present **COMBRL**, a continuous-time, optimistic model-based reinforcement learning (MBRL) algorithm. **COMBRL** proceeds in a continuous-time, episodic setting, alternating between learning a predictive model of the dynamics from data and selecting policies that trade off extrinsic reward and epistemic uncertainty. The method assumes only access to a simulator or physical system for episodic rollouts and measurements.

### 3.1 OPTIMISTIC PLANNING OBJECTIVE

In each episode $n$, we select *any* $L_f$-Lipschitz model from the confidence set $\mathcal{M}_{n-1}$ of the previous episode, i.e., $\boldsymbol{f}_n \in \mathcal{M}_{n-1} \cap \mathcal{F}$, where $\mathcal{F}$ is the set of $L_f$-Lipschitz functions. We then choose a policy $\boldsymbol{\pi}_n$ that maximizes a reward-augmented optimistic objective under the current model $\boldsymbol{f}_n$:

$$\boldsymbol{\pi}_n = \arg\max_{\boldsymbol{\pi} \in \Pi} \int_0^T \frac{r(\boldsymbol{x}'(s), \boldsymbol{u}(s)) + \lambda_n \cdot \|\boldsymbol{\sigma}_{n-1}(\boldsymbol{x}'(s), \boldsymbol{u}(s))\|}{1 + \lambda_n} \, ds \tag{3}$$

$$\text{s.t.} \quad \dot{\boldsymbol{x}}'(t) = \boldsymbol{f}_n(\boldsymbol{x}'(t), \boldsymbol{u}(t)), \quad \boldsymbol{u}(t) = \boldsymbol{\pi}(\boldsymbol{x}'(t)).$$

$$(\boldsymbol{x}(t), \boldsymbol{u}(t)) \in \mathcal{Z} = \mathcal{X} \times \mathcal{U} \subset \mathbb{R}^{d_x + d_u}, \quad t \in [0, T].$$

---

**Algorithm 1 COMBRL**: Continuous-Time Optimistic MBRL

---

1: **Initialize:** Statistical model $\mathcal{M}_0$, Simulator $\text{SIM}(\cdot, \cdot)$, Dataset $\mathcal{D}_0 = \emptyset$, measurement selection
    strategy $S = (S_n)_{n \geq 1}$, intrinsic reward weights $(\lambda_n)_{n=1}^N$, confidence level $\delta$
2: **for** episode $n = 1, \ldots, N$ **do**
3:     $\boldsymbol{\pi}_n \leftarrow \text{OPTIMIZEPOLICY}(\mathcal{M}_{n-1}, \lambda_n)$   ➤ Solve optimistic objective from Equation (3),
                                                    subject to $L_f$-Lipschitz dynamics $\boldsymbol{f}_n \in \mathcal{F}$
4:     $\mathcal{D}_n \leftarrow \text{SIM}(\boldsymbol{\pi}_n, S_n)$                       ➤ Collect rollout using $S_n$
5:     $\mathcal{M}_n \leftarrow \text{UPDATEMODEL}(\mathcal{D}_{0:n}, \delta)$     ➤ Fit mean $\boldsymbol{\mu}_n$ and uncertainty $\boldsymbol{\sigma}_n$

---

The key feature of **COMBRL** is its reward-plus-uncertainty objective in continuous time, which enables a principled trade-off between exploration and exploitation. The scalar $\lambda_n$ balances reward and model uncertainty, and is treated as a tunable hyperparameter in practice. The epistemic uncertainty term $\boldsymbol{\sigma}_{n-1}(\boldsymbol{z})$ in Equation (3) encourages the agent to visit poorly understood regions of the state-action space.

Earlier continuous-time optimistic methods addressed exploration by jointly optimizing over policies and dynamics $\boldsymbol{f}_n \in \mathcal{M}_{n-1} \cap \mathcal{F}$. This is intractable and heuristically addressed using a reparametrization trick from Curi et al. (2020), which increases input dimensionality from $d_u$ to $d_u + d_x$, limiting scalability in high-dimensional settings, e.g., control from pixels. **COMBRL** avoids this by selecting *any* model from $\mathcal{M}_{n-1} \cap \mathcal{F}$; in practice, using $\boldsymbol{\mu}_n$ works well.[2] Unlike optimistic dynamics or classical control, **COMBRL** balances exploration and exploitation via a single scalar $\lambda_n$ and encourages exploration of uncertain regions to improve model fidelity. **COMBRL** is summarized in Algorithm 1.

### 3.2 The internal reward weight $\lambda_n$

The scalar value $\lambda_n$ in Equation (3) determines the trade-off between maximizing extrinsic reward and exploring regions of high model uncertainty. We differentiate between three key scenarios that affect the agent's behaviour:

- **Greedy** ($\lambda_n = 0$): Pure exploitation with respect to the given reward function. The model is only updated passively through whatever data results from reward-seeking behaviour, as in prior continuous-time MBRL approaches (Yildiz et al., 2021).

- **Balanced** ($0 < \lambda_n < \infty$): The agent balances task reward and model uncertainty. This regime leads to goal-directed yet exploratory behaviour and is the most relevant in practice, reducing epistemic uncertainty and improving model quality over time. We offer some strategies to select $\lambda_n$ subsequently.

- **Unsupervised** ($\lambda_n \to \infty$): The agent ignores the reward and acts purely to reduce uncertainty. This corresponds to an unsupervised RL or active learning setting. This is similar to some discrete-time methods that use the model disagreement or epistemic uncertainty as an intrinsic reward (Pathak et al., 2019; Sekar et al., 2020; Sukhija et al., 2023).

**How to choose $\lambda_n$ in the balanced case?** For the case $0 < \lambda_n < \infty$, we study several practical strategies for setting or adapting $\lambda_n$:

- **Static (hyperparameter):** Set $\lambda_n = \lambda$ to a fixed value tuned via cross-validation or grid search. This is simple and often effective.

- **Scheduled (annealing):** Decrease $\lambda_n$ over time, for example $\lambda_n = \lambda_0 \cdot (1 - n/N)$. This encourages more exploration early in training and more exploitation as the model improves.

- **Auto-tuned:** Use an information-theoretic approach such as the auto-tuning procedure described by Sukhija et al. (2025a), which selects $\lambda_n$ adaptively based on maximizing mutual information gain or related criteria.

In this work, we primarily treat $\lambda_n$ as a tunable hyperparameter with scheduling. In Section 4, we also study the auto-tuning approach and show its effectiveness, as well as the unsupervised RL case.

---

[2]Even though the mean model might not lie in $\mathcal{M}_{n-1} \cap \mathcal{F}$. For GP dynamics, we show how to pick a model from $\mathcal{M}_{n-1} \cap \mathcal{F}$ in Appendix B.2.

### 3.3 THEORETICAL RESULTS

The regret of any continuous-time model-based RL algorithm depends on the hardness of learning the true dynamics $\boldsymbol{f}^*$ and on the measurement selection strategy (MSS) $S$ from Definition 1. Crucially, in continuous-time RL, measurements can be taken at arbitrary times, with the MSS $S$ specifying when they occur; for example, an *equidistant* MSS chooses uniformly spaced times. For the underlying dynamics $\boldsymbol{f}^*$ and MSS $S$, the model complexity is defined as:

$$\mathcal{I}_N(\boldsymbol{f}^*, S) \stackrel{\text{def}}{=} \max_{\substack{\boldsymbol{\pi}_1, \dots, \boldsymbol{\pi}_N \\ \boldsymbol{\pi}_n \in \Pi}} \sum_{n=1}^{N} \int_0^T \|\boldsymbol{\sigma}_{n-1}(\boldsymbol{z}_n(t))\|^2 \, dt. \tag{4}$$

Intuitively, for a given $N$, the more complicated the dynamics $\boldsymbol{f}^*$, the larger the epistemic uncertainty and thereby the model complexity. Curi et al. (2020) study the model complexity for the discrete-time setting, where the integral is replaced by the sum over uncertainties. In continuous-time, the MSS $S$ proposes when we measure the system and influences how we collect data to update our model.

Unlike discrete time, where uncertainty is assessed on a fixed grid, the continuous-time model complexity integrates uncertainty along the entire trajectory. To compete with the optimal continuous-time policy, one needs increasingly dense observations along trajectories; hence $|S_n|$ must increase with $n$. For example, Treven et al. (2023) show that for an equidistant MSS with $|S_n| = n$, the model complexity is bounded by

$$\mathcal{I}_N(\boldsymbol{f}^*, S) = \mathcal{O}(\gamma_N + \log N),$$

where $\gamma_N$ is the maximum information gain of $N$ noisy observations under the chosen kernel. This growth condition is necessary because, asymptotically, recovering the full ODE requires ever denser coverage of the trajectory, while with a fixed per-episode budget the uncertainty integral cannot shrink quickly enough. With this intuition in place, we can now formalize how the regret of **COMBRL** scales with the model complexity.

**Theorem 1.** *Under regularity assumptions (Assumptions 1 to 3), the following holds with a probability of at least $1 - \delta$:*

$$R_N \leq \mathcal{O}\left(\sqrt{\mathcal{I}_N^3(\boldsymbol{f}^*, S)\, N}\right).$$

Theorem 1 bounds the regret of **COMBRL** w.r.t. the model complexity. Importantly, this shows that the learning performance depends also on how efficiently the MSS collects informative measurements.

For GP dynamics, where the well-calibration assumption is true and the monotonicity of the variance holds, Treven et al. (2023) show that the model complexity $\mathcal{I}_N(\boldsymbol{f}^*, S)$ is sublinear for many common kernels and MSSs, e.g., grows with $\text{poly} \log(N)$ for the RBF kernel and equidistant MSS. Therefore, for common kernels and MSSs, **COMBRL** enjoys sublinear regret in the GP setting, and the policy converges to the optimal policy $\boldsymbol{\pi}^*$.

For the unsupervised setting without an external reward signal, the regret term as defined in Equation (2) does not apply, and instead we analyze the sample complexity of reducing epistemic uncertainty.

**Theorem 2.** *Consider the unsupervised setting ($\lambda_n \to \infty$) and let Assumptions 1 to 3 hold. If $\sigma_{n-1,j}(\boldsymbol{z}) \geq \sigma_{n,j}(\boldsymbol{z}) \ \forall \ \boldsymbol{z} \in \mathcal{Z}, j \leq d_x$, and $n > 0$, then*

$$\max_{\boldsymbol{\pi} \in \Pi} \int_0^T \|\boldsymbol{\sigma}_{n-1}(\boldsymbol{x}(s), \boldsymbol{\pi}(\boldsymbol{x}(s)))\| \, ds \leq \mathcal{O}\left(\sqrt{\frac{\mathcal{I}_N^3(\boldsymbol{f}^*, S)}{N}}\right).$$

Theorem 2 provides a sample complexity bound for the unsupervised case. Effectively, this shows that pure intrinsic exploration with $\lambda_n \to \infty$ reduces our model epistemic uncertainty with a rate of $\sqrt{\mathcal{I}_N^3/N}$. To the best of our knowledge, we are the first to show this for continuous-time RL.

We provide the proofs for all our theoretical results in Appendix B.

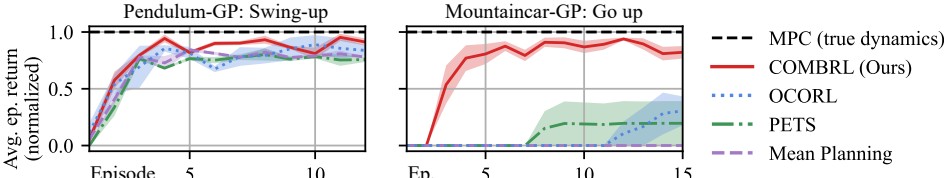

Figure 1: *GP dynamics.* Learning curves for baselines, **COMBRL** and OCORL with fixed internal reward weight $\lambda_n$ using GP dynamics and iCEM planning, averaged over 5 seeds. We report the mean and the standard error bands, and additionally the performance of MPC under the true dynamics as an estimate for the performance of the optimal policy. **COMBRL** achieves higher asymptotic returns than PETS and mean, while matching or exceeding OCORL at roughly $3\times$ lower computational cost (Appendix C.2).

## 4 EXPERIMENTS

We evaluate the **COMBRL** algorithm on various environments from the OpenAI/Farama gymnasium benchmark suite (Gym, Brockman et al., 2016; Towers et al., 2024) and the DeepMind control suite (DMC, Tassa et al., 2018; Tunyasuvunakool et al., 2020). For the dynamics model $\boldsymbol{f}_n$, we use Gaussian processes (GPs) and probabilistic ensembles (PEs) to capture uncertainty about well-calibrated statistical models. Since **COMBRL** models continuous-time dynamics directly, it also remains agnostic to the solver or discretization scheme, and can in principle accommodate adaptive strategies.

Earlier work has shown that continuous-time formulations can outperform discrete-time counterparts in sample efficiency. Building on this, our experiments are designed to show both *(i)* the core behavior of **COMBRL** under standard equidistant sampling and fixed-rate control, and *(ii)* that in principle **COMBRL** can also be used in the time-adaptive setting, where it requires fewer interactions.

Concretely, we first evaluate **COMBRL** with equidistant MSS and fixed control rates, solving the continuous-time planning problem using discrete-time solvers with fine-grained discretization to ensure accurate approximation of the underlying ODE. Unlike discrete-time RL, however, $\boldsymbol{f}_n$ learns the ODE of the true system $\boldsymbol{f}^*$. We then extend our evaluation to the time-adaptive setting, comparing adaptive MSS against fixed-rate control.

We use $\boldsymbol{f}_n$ to generate simulated trajectories for policy training using soft actor-critic (SAC, Haarnoja et al., 2018) for closed-loop control and the improved cross-entropy method (iCEM, Pinneri et al., 2021) for real-time trajectory optimization. For the time-adaptive control experiments, we additionally employ the time-adaptive control & sensing (TaCoS) framework of Treven et al. (2024) as an alternative planner.

Brief descriptions of SAC and iCEM as well as additional implementation details are provided in Appendix C.1 and C.4, while further details on the TaCoS framework are given in Appendix C.7.

**Baselines** We compare **COMBRL** with two baselines with different planning approaches. The mean planner uses the mean estimate $\boldsymbol{\mu}_n$ of the statistical model and greedily maximizes the extrinsic reward for the task at hand (i.e., $\lambda_n = 0$), akin to Yildiz et al. (2021).
Furthermore, we compare our method with the trajectory sampling scheme (TS-1) proposed by Chua et al. (2018), subsequently referred to as PETS. PETS samples trajectories from PEs for state propagation and thus inherently captures the epistemic uncertainty during planning.

**Does COMBRL scale better than the state of the art?** In Figure 1, we empirically validate our theoretical insights using GPs. We consider two classic continuous control tasks – the pendulum swing-up and the mountaincar problem (Moore, 1990) – and use iCEM for real-time planning. We compare **COMBRL** with the baselines above and with the state-of-the-art continuous-time RL algorithm OCORL from Treven et al. (2023). In these experiments, $\lambda_n$ is held constant throughout training and treated as a static, hand-tuned hyperparameter.

After each episode, we evaluate the learned model by computing the return using its mean prediction $\boldsymbol{\mu}_n$ on the original task. As a lower bound for the performance of the optimal policy $\boldsymbol{\pi}^*$, we report the best performance achieved by MPC (iCEM) on the true system when the dynamical system $\boldsymbol{f}^*$ is known. Thus, the MPC can plan with respect to the task/reward at hand subject to the true dynamics.

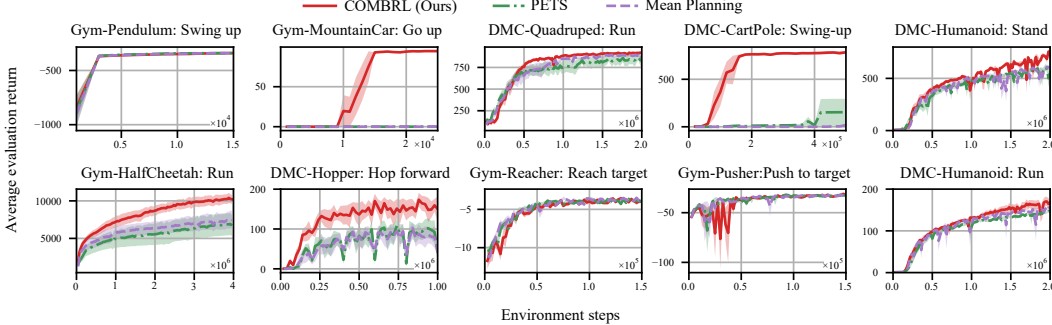

Figure 2: *Effect of intrinsic rewards.* Learning curves for **COMBRL** (with auto-tuned $\lambda_n$) and baselines on different continuous control tasks. We report the mean return when evaluating the learned model on the task at hand, averaged across 10 random seeds along with the standard error. **COMBRL** achieves the largest performance gains in sparse or underactuated tasks, and consistent improvements in higher-dimensional domains.

Results, averaged over five random seeds with standard error bands, show that **COMBRL** consistently achieves higher asymptotic returns than the baselines, confirming the benefits of intrinsic rewards for guiding exploration. For the Pendulum experiments, all algorithms solve the task of swinging up. However, the co-optimization over the reward and the optimistic dynamics as well as the reparametrization trick used by the OCORL algorithm are computationally prohibitive, and require around $3\times$ the compute time compared to **COMBRL** (see Appendix C.2). Thus our algorithm scales better and is more computationally efficient than the state of the art.

**How does the intrinsic reward affect learning?** To assess the effect of intrinsic rewards, we compare **COMBRL** with $\lambda_n > 0$ against PETS and the mean planner. Figure 2 shows learning curves on several environments from Gym and DMC. In these experiments, we model dynamics using PEs and use SAC to train the policy. We compare **COMBRL** with a nonzero intrinsic reward weight $\lambda_n$ against PETS and the mean planner ($\lambda_n = 0$). For **COMBRL**, the internal reward weight $\lambda_n$ is automatically tuned following the strategy proposed by Sukhija et al. (2025a). Details of the tuning procedure and the resulting evolution of $\lambda_n$ over training are provided in Appendix C.3. We further show in Appendix C.6 that this auto-tuned intrinsic reward drives exploration towards epistemically uncertain regions of the state–action space.

Across a range of environments, **COMBRL** achieves higher asymptotic returns, particularly in sparse-reward or underactuated settings such as MountainCar and CartPole, highlighting that optimism-driven exploration significantly accelerates learning. In higher-dimensional environments such as HalfCheetah, Hopper, and Humanoid, COMBRL improves performance by encouraging exploration of uncertain regions, which helps uncover effective behaviors in complex, high-degree-of-freedom systems, even when these behaviors are not directly tied to high immediate reward signals.

**How does COMBRL perform in the unsupervised RL setting?** We evaluate whether exploration driven by model uncertainty improves generalization to unseen tasks. Specifically, we train policies on a primary task (e.g., reaching a target) and assess zero-shot performance on a downstream task not encountered during training (e.g., moving away from the target). The downstream tasks are implemented by modifying the reward per environment and are summarized in Appendix C.5.

Figure 3 compares standard **COMBRL** as well as its unsupervised variant ($\lambda_n \to \infty$, cf. Section 3.2) to the baselines. To ensure a fair comparison, we let each agent explore the environment for several episodes, and then periodically evaluate the learned model on downstream tasks. While **COMBRL** generally achieves the best performance on the primary task especially in environments which offer sparse rewards, its unsupervised variant performs best on the downstream task across all seven evaluated domains.

This suggests that exploration guided by model uncertainty encourages the agent to cover a more diverse state-action space and highlights the trade-off between task focus and exploration breadth: Ignoring the reward signal entirely leads the agent to explore broadly, acquiring a globally accurate model that generalizes better to unseen tasks for $\lambda_n \to \infty$.

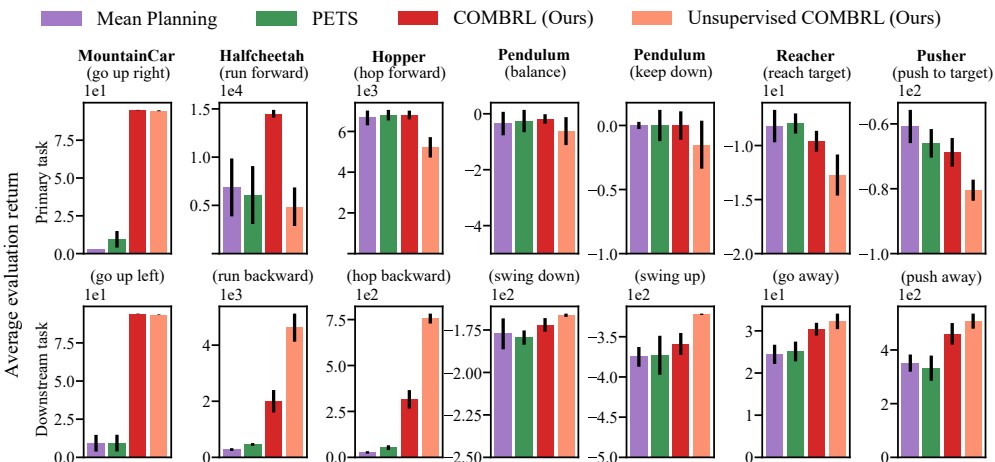

Figure 3: *Generalization to downstream tasks.* Final returns at convergence on primary (trained) and downstream (unseen) tasks across seven Gym environments. We report the mean return as well as the standard error for the primary and downstream task. For **COMBRL**, we differentiate between the balanced case with a static or annealing schedule for $\lambda_n$, or the unsupervised case with $\lambda_n \to \infty$.

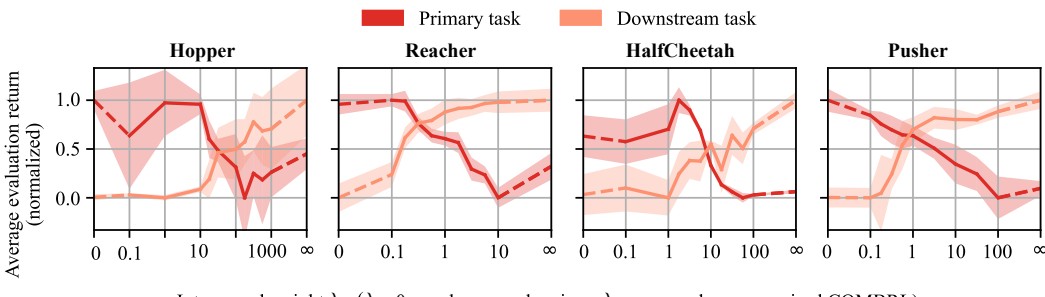

Figure 4: *Ablating the internal reward weight $\lambda_n$.* Final performance at convergence for different environments and tasks with varying $\lambda$. We ablate over different choices of $\lambda$ and report the mean return and standard error on a primary task which the proposed algorithm was trained on, as well as a previously unseen downstream task.

**How does the choice of $\lambda_n$ trade off directed exploration w.r.t. the extrinsic reward and global exploration?** We ablate different values of the internal reward weight $\lambda_n$ for the Gym implementation of the HalfCheetah (Wawrzyński, 2009), Hopper (Erez et al., 2012), as well as the Reacher and Pusher, which are part of the MuJoCo environments (Todorov et al., 2012). Figure 4 shows that growing nonzero $\lambda_n$ values improve downstream generalization while maintaining strong performance on the primary task. In contrast, large $\lambda_n$ may overly favor exploration and degrade performance on the primary task. This suggests that there exists an intermediate value of $\lambda_n$ that balances goal-directed behaviour with model uncertainty reduction, achieving the best of both **COMBRL** (task-optimal) and its unsupervised variant (exploration-optimal).

**How does COMBRL perform in the time-adaptive setting?** A key advantage of continuous-time RL over its discrete-time counterpart is that it enables us to deploy policies at varying control frequencies. To illustrate this benefit, we extend our evaluation to the time-adaptive control & sensing (TaCoS) framework from Treven et al. (2024), where interactions with the system incur transition costs and the agent jointly optimizes both actions and their durations, i.e., the control frequency. We offer details on the TaCoS framework in Appendix C.7.

We compare **COMBRL** with time-adaptive MSS (**COMBRL**-TaCoS) with OTaCoS (the variant of OCORL for TaCoS), Mean-TaCoS, and PETS-TaCoS. Furthermore, we compare these time-adaptive methods to COMBRL with a fixed control rate and equidistant MSS such as the one used in the previous experiments. Across benchmarks, **COMBRL**-TaCoS achieves competitive or superior returns while using fewer interactions compared to the fixed control rate variant, which is shown by

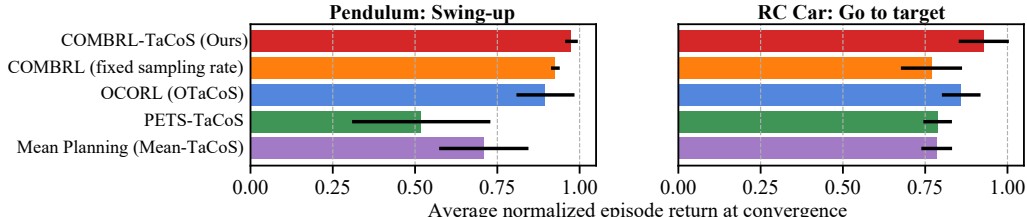

Figure 5: *Adaptive TaCoS setting.* Final performance at convergence for **COMBRL**-TaCoS compared to OTaCoS, Mean-TaCoS, PETS-TaCoS, and COMBRL with a fixed control rate (equidistant MSS). Final returns at convergence are averaged over 10 random seeds and reported as mean with standard error over 10 random seeds. **COMBRL**-TaCoS achieves competitive or superior returns while requiring fewer interactions than its fixed-rate variant, and matches or exceeds the performance of OTaCoS.

the higher returns in Figure 5 and in turn, demonstrates sample efficiency. Compared to the previous experiments, this shows that optimism-driven exploration extends naturally to the adaptive-control setting and that **COMBRL** is, in principle, applicable independently of not only the dynamics model, but also the measurement selection strategy.

## 5 CONCLUSION

In this work, we introduced **COMBRL**, a continuous-time model-based reinforcement learning algorithm that uses epistemic uncertainty to guide exploration through an intrinsic reward. Our approach provides a principled and flexible mechanism to balance exploration and exploitation via the internal reward weight $\lambda_n$, generalizing the classical optimism-in-the-face-of-uncertainty paradigm to continuous-time domains in a way that is scalable, easy to implement, and agnostic to the statistical model, trajectory planner, as well as the measurement and control strategy.

Our experiments demonstrate that **COMBRL** is strong at goal-directed learning on the task at hand, while its unsupervised RL variant (i.e., $\lambda_n \to \infty$) is particularly effective at generalizing to unseen downstream tasks. This highlights a core trade-off: Reward directed exploration for appropriate $\lambda_n$ values and exploration across the entire reachable domain for $\lambda_n \to \infty$. We differentiate between three cases for the internal reward weight that affect the agent's behaviour – greedy, balanced, and unsupervised. Empirical ablations confirm that there exists a range for $\lambda_n$ that enables generalizable yet sample-efficient learning. Furthermore, the unsupervised version of **COMBRL** acts as an unsupervised system identification strategy, enabling strong zero-shot adaptation to new objectives.

### LLM USAGE STATEMENT

Parts of this text were revised with the assistance of a large language model to aid or polish writing and to improve grammar and clarity; the authors remain responsible for all content.

### REPRODUCIBILITY STATEMENT

All algorithmic details of **COMBRL**, including the problem formulation, regret analysis, and assumptions, are presented in the main text, with complete proofs provided in Appendix B. The experimental setup, including environment details, model architectures, and hyperparameters, is described in Appendix C. The source code for the conducted experiments is available under `https://github.com/lasgroup/ombrl`.

### ACKNOWLEDGEMENTS

We would like to thank Bruce D. Lee for the insightful feedback on this work. This project has received funding from the Swiss National Science Foundation under NCCR Automation, grant agreement 51NF40 180545. Numerical simulations were performed on the ETH Zürich Euler cluster.

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

APPENDICES AND SUPPLEMENTARY MATERIAL

# A  ADDITIONAL RELATED WORK

## A.1  MODEL-BASED REINFORCEMENT LEARNING

Model-based RL (MBRL) has emerged as a sample-efficient alternative to model-free methods, with applications ranging from robotics to online decision-making (Chua et al., 2018; Janner et al., 2019; Hansen et al., 2023; Rothfuss et al., 2024). Recent deep MBRL methods differ primarily in dynamics modeling and planning strategies, yet often rely on naive exploration heuristics such as Boltzmann exploration (Hansen et al., 2024; Hafner et al., 2025). However, such heuristics are suboptimal even in simple settings (Cesa-Bianchi et al., 2017).

**COMBRL** addresses this by introducing a principled exploration mechanism that combines epistemic uncertainty with extrinsic reward. Unlike prior methods, it is model- and planner-agnostic, scalable, and comes with sublinear regret guarantees. We show that this intrinsic reward formulation not only improves theoretical performance but also enables meaningful exploration across deep RL benchmarks, where naive methods fail.

## A.2  UNSUPERVISED REINFORCEMENT LEARNING AND INTRINSIC EXPLORATION

System identification (Åström and Eykhoff, 1971) and unsupervised exploration (Aubret et al., 2019) require efficient exploration, as the agent must learn accurate global dynamics models without access to extrinsic rewards. In such settings, exploration is essential for covering informative regions of the state-action space. To this end, intrinsic motivation techniques have long been used to encourage exploration in RL, making use of objectives like model prediction error (Pathak et al., 2017), novelty (Bellemare et al., 2016), empowerment (Salge et al., 2014), and information gain (Sekar et al., 2020; Sukhija et al., 2023). However, such techniques are often used in isolation from extrinsic rewards. **COMBRL** instead combines epistemic uncertainty, which poses a principled intrinsic signal, with task rewards, aligning exploration with learning progress.

In the unsupervised setting, **COMBRL** reduces to active learning Taylor et al. (2021), specifically uncertainty sampling (Lewis and Catlett, 1994) and guides the agent toward regions where the model is most uncertain. Similar ideas have been explored in bandits (Auer et al., 2002; Srinivas et al., 2012), data-driven control (Grimaldi et al., 2024), and RL (Abeille and Lazaric, 2020; Sukhija et al., 2025a), where joint optimization of reward and model uncertainty is shown to improve learning. For the case of data-driven control in linear-quadratic systems, Zhao et al. (2025a;b) use a similar weighting mechanism which is initially motivated by robustness and regularization, but interpretable as trading off exploitation and exploration, showing how uncertainty-aware regularization or exploration bonuses can systematically balance performance and robustness in continuous control.

**COMBRL** follows this direction, using model epistemic uncertainty as a reward bonus to guide exploration. Our work is closely related to Sukhija et al. (2025b), who propose a similar reward formulation in the discrete-time setting. However, our focus on continuous-time systems leads to significantly different theoretical analysis and experimental design. In particular, we derive regret and sample complexity bounds tailored to the continuous-time domain and analyze the effect of the intrinsic reward weight $\lambda_n$ in greater depth. Unlike prior work, which is mostly empirical or limited to linear systems, **COMBRL** also provides theoretical guarantees for general nonlinear systems in continuous time and demonstrates scalability to high-dimensional tasks.

## A.3  CONTINUOUS-TIME REINFORCEMENT LEARNING

While most model-based RL methods are developed in discrete time, continuous-time formulations have gained increasing interest due to their relevance for real-world control and physical modelling (Doya, 2000; Frémaux et al., 2013; Yildiz et al., 2021). Recent works explore learning dynamics via neural ODEs (Chen et al., 2018) and physics-informed priors (Greydanus et al., 2019; Cranmer et al., 2020).

Yildiz et al. (2021) propose a continuous-time actor-critic method that plans using the posterior mean of the learned ODE model. Treven et al. (2023; 2024) derive regret bounds for continuous-time MBRL using optimistic dynamics and highlight the role of different measurement selection strategies, showing that adaptive sampling can substantially reduce the number of interactions needed to match or surpass discrete-time methods. A complementary line of research is event- and self-triggered control (Åström and Bernhardsson, 2002; Anta and Tabuada, 2010; Heemels et al., 2012; 2021),

which seeks efficiency by updating control inputs only when certain conditions are met, thereby avoiding unnecessary interventions while maintaining stability.

**COMBRL** extends this line of work by proposing a flexible framework for both reward-driven and unsupervised exploration in continuous time. In contrast to prior methods, **COMBRL** incorporates optimism directly in the reward function and thus offers a simple, scalable, and theoretically grounded approach that operates directly in the continuous-time domain and supports general-purpose dynamics models.

## B    Theory

We provide the assumptions and proofs for Theorems 1 and 2, which formalize the regret and exploration guarantees of **COMBRL**, in this section. The former bounds regret in terms of model complexity, which is sublinear for common GP kernels and MSSs, implying convergence to the optimal policy. The latter shows that intrinsic exploration alone (i.e., $\lambda \to \infty$) reduces epistemic uncertainty at a rate of $\sqrt{\mathcal{I}_N^3/N}$. To the best of our knowledge, we are the first to show this for continuous-time RL.

### B.1    Assumptions

In the following, we restate the assumptions from Section 2.3 for completeness. We make some common assumptions (cf. Curi et al. (2020); Treven et al. (2023)) that allow us to theoretically analyse the regret $R_N$ and prove a regret bound. We first make an assumption on the continuity of the underlying system and the observation noise.

**Restated Assumption 1** (Lipschitz continuity). *The dynamics model $\boldsymbol{f}^*$, reward $r$, and all policies $\boldsymbol{\pi} \in \Pi$ are $L_f$, $L_r$ and $L_{\boldsymbol{\pi}}$ Lipschitz-continuous, respectively.*

**Restated Assumption 2** (Sub-Gaussian noise). *We assume that the measurement noise $\epsilon_{n,i}$ is i.i.d. $\sigma$-sub Gaussian.*

The Lipschitz assumption is commonly made for analysing nonlinear systems (Khalil, 2014) and is satisfied for many real-world applications. Furthermore, assuming $\sigma$-sub Gaussian noise (Rigollet and Hütter, 2023) is also fairly general and is common in both RL and Bayesian optimization literature (Srinivas et al., 2012; Chowdhury and Gopalan, 2017; Curi et al., 2020).

In **COMBRL**, we learn an uncertainty-aware model of the underlying dynamics. Therefore, we obtain a mean estimate $\boldsymbol{\mu}_n(\boldsymbol{z})$ and quantify our epistemic uncertainty $\boldsymbol{\sigma}_n(\boldsymbol{z})$ about the function $\boldsymbol{f}^*$.

**Restated Definition 2** (Well-calibrated statistical model of $\boldsymbol{f}^*$, Rothfuss et al. (2023)). *Let $\mathcal{Z} = \mathcal{X} \times \mathcal{U}$. An all-time well-calibrated statistical model of the function $\boldsymbol{f}^*$ is a sequence $\{\mathcal{M}_n(\delta)\}_{n \geq 0}$, where*

$$\mathcal{M}_n(\delta) \stackrel{def}{=} \left\{ \boldsymbol{f} : \mathcal{Z} \to \mathbb{R}^{d_x} \mid \forall \boldsymbol{z} \in \mathcal{Z}, \forall j \in \{1, \ldots, d_x\} : |\mu_{n,j}(\boldsymbol{z}) - f_j(\boldsymbol{z})| \leq \beta_n(\delta)\sigma_{n,j}(\boldsymbol{z}) \right\},$$

*if, with probability at least $1 - \delta$, we have $\boldsymbol{f}^* \in \bigcap_{n \geq 0} \mathcal{M}_n(\delta)$. Here, $\mu_{n,j}$ and $\sigma_{n,j}$ denote the $j$-th element in the vector-valued mean and standard deviation functions $\boldsymbol{\mu}_n$ and $\boldsymbol{\sigma}_n$ respectively, and $\beta_n(\delta) \in \mathbb{R}_{\geq 0}$ is a scalar function that depends on the confidence level $\delta \in (0, 1]$ and which is monotonically increasing in $n$.*

**Restated Assumption 3** (Well-calibration of the model). *The learned model is an all-time well-calibrated statistical model of $\boldsymbol{f}^*$, i.e., with probability at least $1 - \delta$, we have $\boldsymbol{f}^* \in \bigcap_{n \geq 0} \mathcal{M}_n(\delta)$ for confidence sets $\mathcal{M}_n(\delta)$ as defined in Definition 2. Moreover, the standard deviation functions $\boldsymbol{\sigma}_n : \mathcal{Z} \to \mathbb{R}^{d_x}$ are $L_{\boldsymbol{\sigma}}$-Lipschitz continuous for all $n \geq 0$.*

Intuitively, Assumption 3 states that we are, with high probability, able to capture the dynamics within a confidence set spanned by our predicted mean and epistemic uncertainty. For Gaussian process (GP) models, the assumption is satisfied (Rothfuss et al., 2023, Lemma 3.6) and for more general classes of models such as Bayesian neural networks (BNNs), re-calibration techniques (Kuleshov et al., 2018) can be used.

Lastly, we make an assumption on the regularity of the dynamics by placing them in a reproducing kernel Hilbert space (RKHS):

**Restated Assumption 4** (RKHS Prior on Dynamics). *We assume that the functions $f_j^*$, $j \in \{1, \ldots, d_{\boldsymbol{x}}\}$ lie in a RKHS with kernel $k$ and have a bounded norm $B$, that is*

$$\boldsymbol{f}^* \in \mathcal{H}_{k,B}^{d_{\boldsymbol{x}}}, \quad with \quad \mathcal{H}_{k,B}^{d_{\boldsymbol{x}}} = \{\boldsymbol{f} \mid \|f_j\|_k \leq B, j = 1, \ldots, d_{\boldsymbol{x}}\}.$$

*Moreover, we assume that $k(\boldsymbol{z}, \boldsymbol{z}) \leq \sigma_{\max}$ for all $\boldsymbol{x} \in \mathcal{X}$.*

## B.2 ANALYSIS OF GAUSSIAN PROCESS DYNAMICS

Assumption 4 allows us to model $\boldsymbol{f}^*$ with GPs. The posterior mean $\boldsymbol{\mu}_n(\boldsymbol{z}) = [\mu_{n,j}(\boldsymbol{z})]_{j \leq d_{\boldsymbol{x}}}$ and epistemic uncertainty $\boldsymbol{\sigma}_n(\boldsymbol{z}) = [\sigma_{n,j}(\boldsymbol{z})]_{j \leq d_{\boldsymbol{x}}}$ can then be obtained using

$$\begin{aligned}
\mu_{n,j}(\boldsymbol{z}) &= \boldsymbol{k}_n^\top(\boldsymbol{z})(\boldsymbol{K}_n + \sigma^2 \boldsymbol{I})^{-1} \boldsymbol{y}_{1:n}^j, \\
\sigma_{n,j}^2(\boldsymbol{z}) &= k(\boldsymbol{z}, \boldsymbol{z}) - \boldsymbol{k}_n^\top(\boldsymbol{z})(\boldsymbol{K}_n + \sigma^2 \boldsymbol{I})^{-1} \boldsymbol{k}_n(\boldsymbol{z}),
\end{aligned} \tag{5}$$

Here, $\boldsymbol{y}_{1:n}^j$ corresponds to the noisy measurements of $f_j^*$, i.e., the noisy derivative observation from the dataset $\mathcal{D}_{1:n}$, $\boldsymbol{k}_n(\boldsymbol{z}) = [k(\boldsymbol{z}, \boldsymbol{z}_i)]_{\boldsymbol{z}_i \in \mathcal{D}_{1:n}}$, and $\boldsymbol{K}_n = [k(\boldsymbol{z}_i, \boldsymbol{z}_l)]_{\boldsymbol{z}_i, \boldsymbol{z}_l \in \mathcal{D}_{1:n}}$ is the data kernel matrix.

Moreover, since $\boldsymbol{f}^*$ has bounded RKHS norm, i.e., $\|\boldsymbol{f}^*\|_k \leq B$ (Assumption 4), it follows from Srinivas et al. (2012); Chowdhury and Gopalan (2017) that with probability $1 - \delta$ we have for every episode $n$:

$$\|\boldsymbol{f}^* - \boldsymbol{\mu}_n\|_{k_n} \leq \beta_n.$$

Instead of planning with the mean, which in general might not be Lipschitz continuous for all $n$, we select a function $\boldsymbol{f}_n$ that not only approximates the $\boldsymbol{f}^*$ function well, i.e., satisfies $\|\boldsymbol{f}^* - \boldsymbol{f}_n\|_{k_n} \leq \beta_n$, but also has an RKHS norm that does not grow with $n$. To achieve this, we propose solving the following quadratic optimization problem:

$$\boldsymbol{f}_n = \underset{\boldsymbol{f} \in \text{span}(k(\boldsymbol{x}_1, \cdot), \ldots, k(\boldsymbol{x}_n, \cdot))}{\arg \min} \|\boldsymbol{f} - \boldsymbol{\mu}_n\|_{k_n} \tag{6}$$
$$\text{s.t. } \|\boldsymbol{f}\|_k \leq B$$

**Theorem 3.** *The optimization problem Equation (6) is feasible and we have $\|\boldsymbol{f}_n - \boldsymbol{\mu}_n\|_{k_n} \leq 2\beta_n$.*

*Proof.* Consider the noise-free case, i.e., $\epsilon_{n,i} = 0$, and let $\bar{\boldsymbol{\mu}}_n$ be the posterior mean for this setting. For the function $\bar{\boldsymbol{\mu}}_n$, it holds that $\|\boldsymbol{f}^* - \bar{\boldsymbol{\mu}}_n\|_{k_n} \leq \beta_n$ (Kanagawa et al., 2018, Corollary 3.11) and $\|\bar{\boldsymbol{\mu}}_n\|_k \leq B$ (Kanagawa et al., 2018, Theorem 3.5). Thus it follows that

$$\|\bar{\boldsymbol{\mu}}_n - \boldsymbol{\mu}_n\|_{k_n} \leq \|\bar{\boldsymbol{\mu}}_n - \boldsymbol{f}^*\|_{k_n} + \|\boldsymbol{f}^* - \boldsymbol{\mu}_n\|_{k_n} \leq 2\beta_n.$$

By representer theorem, it also holds that $\bar{\boldsymbol{\mu}}_n \in \text{span}(k(\boldsymbol{z}_1, \cdot), \ldots, k(\boldsymbol{z}_n, \cdot))$. $\square$

Let $\boldsymbol{\alpha}_n = (\boldsymbol{K} + \sigma^2 \boldsymbol{I})^{-1} \boldsymbol{y} \in \mathbb{R}^n$ and reparametrize $\boldsymbol{f}(\boldsymbol{x}) = \sum_{i=1}^n \alpha_i k(\boldsymbol{x}_i, \boldsymbol{x})$. We have $\|\boldsymbol{f}\|_k^2 = \boldsymbol{\alpha}^\top \boldsymbol{K} \boldsymbol{\alpha}$. We also have:

$$\|\boldsymbol{f} - \boldsymbol{\mu}_n\|_{k_n}^2 = (\boldsymbol{\alpha} - \boldsymbol{\alpha}_n)^\top \boldsymbol{K} \left(\boldsymbol{I} + \frac{1}{\sigma^2} \boldsymbol{K}\right)(\boldsymbol{\alpha} - \boldsymbol{\alpha}_n)$$

Hence the optimization problem in Equation (6) is equivalent to:

$$\min_{\boldsymbol{\alpha} \in \mathbb{R}^n} (\boldsymbol{\alpha} - \boldsymbol{\alpha}_n)^\top \boldsymbol{K} \left(\boldsymbol{I} + \frac{1}{\sigma^2} \boldsymbol{K}\right)(\boldsymbol{\alpha} - \boldsymbol{\alpha}_n)$$
$$\text{s.t. } \boldsymbol{\alpha}^\top \boldsymbol{K} \boldsymbol{\alpha} \leq B^2$$

This is a quadratic program that can be solved using any QP solver. The program finds the closest function to the posterior mean $\boldsymbol{\mu}_n$ that is Lipschitz continuous. In particular, note that since $\|\boldsymbol{f}_n\|_k \leq B$, for Lipschitz kernels, $\boldsymbol{f}_n$ has a Lipschitz constant $L_B$ which is independent of $n$ (Berkenkamp, 2019). From hereon, let $L_* = \max\{L_f, L_B\}$.

Next, we plan with the dynamics $\boldsymbol{f}_n$ that are obtained from solving Equation (6), i.e.,

$$\boldsymbol{\pi}_n = \arg\max_{\boldsymbol{\pi}\in\Pi} \mathbb{E}_{\boldsymbol{\pi}}\left[\int_0^T \left(r(\boldsymbol{x}'(t), \boldsymbol{u}(t)) + \lambda_n \|\boldsymbol{\sigma}_n(\boldsymbol{x}'(t), \boldsymbol{u}(t))\|\right)dt\right] \tag{7}$$

$$\text{s.t.} \quad \dot{\boldsymbol{x}}'(t) = \boldsymbol{f}_n(\boldsymbol{x}'(t), \boldsymbol{u}(t)).$$

**Lemma 4.** *Let Assumption 2 and Assumption 4 hold. Consider the following definitions:*

$$J(\boldsymbol{\pi}, \boldsymbol{f}^*) = \mathbb{E}\left[\int_0^T r(\boldsymbol{x}(t), \boldsymbol{\pi}(\boldsymbol{x}(t)))\, dt.\right]$$

$$\text{s.t.} \quad \dot{\boldsymbol{x}} = \boldsymbol{f}^*(\boldsymbol{x}(t), \boldsymbol{\pi}(\boldsymbol{x}(t))), \quad \boldsymbol{x}_0 = \boldsymbol{x}(0),$$

$$J(\boldsymbol{\pi}, \boldsymbol{f}_n) = \mathbb{E}\left[\int_0^T r(\boldsymbol{x}'(t), \boldsymbol{\pi}(\boldsymbol{x}'(t)))\, dt.\right]$$

$$\text{s.t.} \quad \dot{\boldsymbol{x}}' = \boldsymbol{f}_n(\boldsymbol{x}'(t), \boldsymbol{\pi}(\boldsymbol{x}'(t))), \quad \boldsymbol{x}_0' = \boldsymbol{x}(0),$$

$$\Sigma_n(\boldsymbol{\pi}, \boldsymbol{f}^*) = \mathbb{E}\left[\int_0^T \|\boldsymbol{\sigma}_n(\boldsymbol{x}(t), \boldsymbol{\pi}(\boldsymbol{x}(t)))\|\, dt.\right]$$

$$\text{s.t.} \quad \dot{\boldsymbol{x}} = \boldsymbol{f}^*(\boldsymbol{x}(t), \boldsymbol{\pi}(\boldsymbol{x}(t))) \quad \boldsymbol{x}_0 = \boldsymbol{x}(0),$$

$$\Sigma_n(\boldsymbol{\pi}, \boldsymbol{f}_n) = \mathbb{E}\left[\int_0^T \|\boldsymbol{\sigma}_n(\boldsymbol{x}'(t), \boldsymbol{\pi}(\boldsymbol{x}'(t)))\|\, dt.\right]$$

$$\text{s.t.} \quad \dot{\boldsymbol{x}}' = \boldsymbol{f}_n(\boldsymbol{x}'(t), \boldsymbol{\pi}(\boldsymbol{x}'(t))), \quad \boldsymbol{x}_0' = \boldsymbol{x}(0).$$

*Furthermore, let $\lambda_n = 2\beta_n L_r(1 + L_{\boldsymbol{\pi}})Te^{L_*(1+L_{\boldsymbol{\pi}})T}$.*

*Then, we have for all $n \geq 0$, $\boldsymbol{\pi} \in \Pi$ with probability at least $1 - \delta$:*

$$|J(\boldsymbol{\pi}, \boldsymbol{f}^*) - J(\boldsymbol{\pi}, \boldsymbol{f}_n)| \leq \lambda_n \Sigma_n(\boldsymbol{\pi}, \boldsymbol{f}_n)$$
$$|J(\boldsymbol{\pi}, \boldsymbol{f}^*) - J(\boldsymbol{\pi}, \boldsymbol{f}_n)| \leq \lambda_n \Sigma_n(\boldsymbol{\pi}, \boldsymbol{f}^*).$$

*Proof.* We follow Treven et al. (2023) and bound the regret with $\Sigma_n(\boldsymbol{\pi}, \boldsymbol{f}_n)$.

$$|J(\boldsymbol{\pi}, \boldsymbol{f}^*) - J(\boldsymbol{\pi}, \boldsymbol{f}_n)| = \mathbb{E}\left[\int_0^T r(\boldsymbol{x}(t), \boldsymbol{\pi}(\boldsymbol{x}(t))) - r(\boldsymbol{x}'(t), \boldsymbol{\pi}(\boldsymbol{x}'(t)))\, dt\right]$$

$$\leq L_r(1 + L_{\boldsymbol{\pi}})\mathbb{E}\left[\int_0^T \|\boldsymbol{x}(t) - \boldsymbol{x}'(t)\|\, dt\right]$$

$$\leq 2\beta_n L_r(1 + L_{\boldsymbol{\pi}})Te^{L_f(1+L_{\boldsymbol{\pi}})T}\int_0^T \|\boldsymbol{\sigma}_{n-1}(\boldsymbol{x}(t), \boldsymbol{\pi}(\boldsymbol{x}(t)))\|\, dt$$

$$\text{(Treven et al. (2023), Lemma 4)}$$

$\square$

**Lemma 5.** *Let Assumption 2 and Assumption 4 hold and consider the simple regret at episode $n$:*

$$r_n = J(\boldsymbol{\pi}^*, \boldsymbol{f}^*) - J(\boldsymbol{\pi}_n, \boldsymbol{f}^*).$$

*The following holds for all $n > 0$ with probability at least $1 - \delta$:*

$$r_n \leq (2\lambda_n + \lambda_n^2)\Sigma_n(\boldsymbol{\pi}_n, \boldsymbol{f}^*).$$

*Proof of Lemma 5.*

$$
\begin{aligned}
r_n &= J(\boldsymbol{\pi}^*, \boldsymbol{f}^*) - J(\boldsymbol{\pi}_n, \boldsymbol{f}^*) \\
&\leq J(\boldsymbol{\pi}^*, \boldsymbol{f}_n) + \lambda_n \Sigma_n(\boldsymbol{\pi}^*, \boldsymbol{f}_n) - J(\boldsymbol{\pi}_n, \boldsymbol{f}^*) && \text{(Lemma 4)} \\
&\leq J(\boldsymbol{\pi}_n, \boldsymbol{f}_n) + \lambda_n \Sigma_n(\boldsymbol{\pi}_n, \boldsymbol{f}_n) - J(\boldsymbol{\pi}_n, \boldsymbol{f}^*) && \text{(Equation (7))} \\
&= J(\boldsymbol{\pi}_n, \boldsymbol{f}_n) - J(\boldsymbol{\pi}_n, \boldsymbol{f}^*) + \lambda_n \Sigma_n(\boldsymbol{\pi}_n, \boldsymbol{f}_n) \\
&\leq \lambda_n \Sigma_n(\boldsymbol{\pi}_n, \boldsymbol{f}^*) + \lambda_n \Sigma_n(\boldsymbol{\pi}_n, \boldsymbol{f}_n) && \text{(Lemma 4)} \\
&= 2\lambda_n \Sigma_n(\boldsymbol{\pi}_n, \boldsymbol{f}^*) + \lambda_n(\Sigma_n(\boldsymbol{\pi}_n, \boldsymbol{f}_n) - \Sigma_n(\boldsymbol{\pi}_n, \boldsymbol{f}^*)) \\
&\leq (\lambda_n^2 + 2\lambda_n)\Sigma_n(\boldsymbol{\pi}_n, \boldsymbol{f}^*).
\end{aligned}
$$

In the last line, we used the fact that $\Sigma_n(\cdot, \cdot)$ is bounded and positive, and thus behaves like a reward. In fact, it is an intrinsic reward (Sukhija et al., 2025b), which is why Lemma 4 also applies.

$\square$

**Restated Theorem 1** (Regret bound in the sub-Gaussian noise case). *Let Assumption 2 and Assumption 4 hold. Then we have for all $N > 0$ with probability at least $1 - \delta$:*

$$
R_N \leq \mathcal{O}\left(\mathcal{I}_N^{3/2}\sqrt{N}\right).
$$

*Proof of Theorem 1.*

$$
\begin{aligned}
R_N &= \sum_{n=1}^N r_n && \text{(Equation (2))} \\
&\leq \sum_{n=1}^N (\lambda_n^2 + 2\lambda_n)\Sigma_n(\boldsymbol{\pi}_n, \boldsymbol{f}^*) && \text{(Lemma 5)} \\
&\leq (\lambda_N^2 + \lambda_N) \sum_{n=1}^N \Sigma_n(\boldsymbol{\pi}_n, \boldsymbol{f}^*) \\
&= (\lambda_N^2 + 2\lambda_N) \sum_{n=1}^N \mathbb{E}_{\boldsymbol{f}^*}\left[\int_0^T \|\boldsymbol{\sigma}_n(\boldsymbol{x}(t), \boldsymbol{\pi}_n(\boldsymbol{x}(t)))\| \, dt\right] \\
&\leq (\lambda_N^2 + 2\lambda_N)\sqrt{NT} \sum_{n=1}^N \mathbb{E}_{\boldsymbol{f}^*}\left[\int_0^T \|\boldsymbol{\sigma}_n^2(\boldsymbol{x}(t), \boldsymbol{\pi}_n(\boldsymbol{x}(t)))\| \, dt\right] \\
&\leq (\lambda_N^2 + 2\lambda_N)\sqrt{NT\mathcal{I}_N(\boldsymbol{f}^*, S)} && \text{(Treven et al. (2023), Proposition 1)}
\end{aligned}
$$

Note that by Lemma 4, $\lambda_N$ scales with the confidence parameter $\beta_N$, which itself depends on the maximum information gain (Chowdhury and Gopalan, 2017; Rothfuss et al., 2023) and thus scales in the model complexity as $\lambda_N^2 = \mathcal{O}(\mathcal{I}_N)$, yielding the result. $\square$

**Restated Theorem 2** (Sample complexity bound in the unsupervised case). *Let Assumption 2 and Assumption 4 hold. Consider Algorithm 1 with extrinsic reward $r = 0$, i.e.,*

$$
\boldsymbol{\pi}_n = \arg\max_{\boldsymbol{\pi} \in \Pi} \mathbb{E}_{\boldsymbol{\pi}}\left[\int_0^{T-1} \|\boldsymbol{\sigma}_n(\boldsymbol{x}'(t), \boldsymbol{\pi}(\boldsymbol{x}'(t)))\| \, dt\right],
$$

$$
s.t. \quad \dot{\boldsymbol{x}}'(t) = \boldsymbol{f}_n(\boldsymbol{x}'(t), \boldsymbol{\pi}(\boldsymbol{x}(t))).
$$

*Then we have for all $N > 0$, with probability at least $1 - \delta$:*

$$
\max_{\boldsymbol{\pi} \in \Pi} \mathbb{E}_{\boldsymbol{f}^*}\left[\int_0^{T-1} \|\boldsymbol{\sigma}_n(\boldsymbol{x}(t), \boldsymbol{\pi}(\boldsymbol{x}(t)))\| \, dt\right] \leq \mathcal{O}\left(\sqrt{\frac{\mathcal{I}_N^3}{N}}\right).
$$

*Proof of Theorem 2.* Recall the definitions from Lemma 4. Let $\Sigma_N^* = \max_{\boldsymbol{\pi}} \Sigma_N(\boldsymbol{\pi}, \boldsymbol{f}^*)$ and let $\boldsymbol{\pi}_N^*$ be the corresponding policy. Recall from Lemma 5 that $\Sigma_n(\cdot, \cdot)$ behaves like a reward.

$$\Sigma_N^* \le \frac{1}{N} \sum_{n=1}^N \Sigma_n^* \qquad \text{(monotonicity of the variance)}$$

$$\le \frac{1}{N} \sum_{n=1}^N (1 + \lambda_n) \Sigma_n(\boldsymbol{\pi}_n^*, \boldsymbol{f}_n) \qquad \text{(Lemma 4)}$$

$$\le \frac{1}{N} \sum_{n=1}^N (1 + \lambda_n) \Sigma_n(\boldsymbol{\pi}_n, \boldsymbol{f}_n) \qquad (\boldsymbol{\pi}_n \text{ is the maximizer for dynamics } \boldsymbol{f}_n)$$

$$\le \frac{1}{N} \sum_{n=1}^N (1 + \lambda_n)^2 \Sigma_n(\boldsymbol{\pi}_n, \boldsymbol{f}^*) \qquad \text{(Lemma 4)}$$

$$\le (1 + \lambda_N)^2 \frac{1}{N} \sum_{n=1}^N \Sigma_n(\boldsymbol{\pi}_n, \boldsymbol{f}^*)$$

$$\le (1 + \lambda_N)^2 \frac{\sqrt{N}}{N} \sqrt{\sum_{n=1}^N \Sigma_n^2(\boldsymbol{\pi}_n, \boldsymbol{f}^*)} \qquad \text{(Cauchy-Schwarz inequality)}$$

$$\le \mathcal{O}\left( \sqrt{\frac{\mathcal{I}_N^3}{N}} \right) \qquad \text{(c.f. Proof of Theorem 1)}$$

$$\square$$

## C  EXPERIMENTAL SETUP

We provide additional details for our experiments in this section.

### C.1  GP EXPERIMENTS

We evaluate our method on two low-dimensional continuous control tasks: Pendulum-GP and MountainCar-GP (Moore, 1990). Unlike in the other, following experiments, these environments are implemented directly by us as continuous-time systems with known physical dynamics given by nonlinear ODEs, rather than relying on Gym or DMC implementations. We emulate a continuous-time setting by using a fine time discretization for state propagation. As for measurements, we assume that we have direct access to the state derivatives.

In **Pendulum-GP**, the agent must swing up and stabilize a pendulum in the upright position. The state vector is defined as

$$\boldsymbol{x} = \begin{bmatrix} x_0 \\ x_1 \\ x_2 \end{bmatrix} = \begin{bmatrix} \cos\theta \\ \sin\theta \\ \dot{\theta} \end{bmatrix}, \quad \boldsymbol{u} = u \in [-2, 2],$$

where $\theta \in [-\pi, \pi]$ is the pendulum angle and $\dot{\theta}$ is the angular velocity. The underlying nonlinear ODE is:

$$\frac{d\theta}{dt} = \dot{\theta}, \quad \frac{d\dot{\theta}}{dt} = \frac{3g}{2\ell} \sin\theta + \frac{3}{m\ell^2} u,$$

with constants $g = 9.81 \text{ m/s}^2$, $m = 1$, and $\ell = 1$. We use a *Gym-style* reward, which penalizes deviations from the target angle $\theta = 0$, angular velocity $\dot{\theta}$, and control input $u$:

$$r(\boldsymbol{x}, \boldsymbol{u}) = -\theta^2 - 0.1\,\dot{\theta}^2 - 0.02\,u^2,$$

where $\theta = \arctan 2(x_1, x_0)$ and $\dot{\theta} = x_2$.

In **MountainCar-GP**, the agent must build momentum to propel a car up a steep hill. The state vector is defined as

$$\boldsymbol{x} = \begin{bmatrix} x_1 \\ x_2 \end{bmatrix}, \quad \boldsymbol{u} = u \in [-1, 1],$$

where $x_1 \in [-1.2, 0.6]$ is the position and $x_2 \in [-0.07, 0.07]$ is the velocity. The underlying nonlinear ODE is given by:

$$\frac{dx_1}{dt} = x_2, \quad \frac{dx_2}{dt} = 0.0015 \cdot u - 0.0025 \cos(3 \cdot x_1).$$

Position and velocity are clipped to their bounds, and backward motion is blocked at $x_1 = -1.2$ if $x_2 < 0$. The reward includes a terminal bonus of +100 for reaching the goal and penalizes control effort:

$$r(\boldsymbol{x}, \boldsymbol{u}) = -0.1\, u^2 + 100 \cdot \mathbf{1}_{\text{goal reached}},$$

where the goal is reached if the car's position exceeds $0.45$ and its velocity is non-negative.

For our GP experiments in Figure 1, we use the RBF kernel. The kernel parameters are updated online using maximum likelihood estimation (Rasmussen and Williams, 2005). We use a hand-tuned static regime for the internal reward weight, i.e., $\lambda_n = \lambda$. For the OCORL baseline, we provide the confidence level function $\beta_n(\delta) = \beta$. The hyperparameters for the statistical model as well as for the environments and algorithms are given in Table 1.

Table 1: Model training hyperparameters and experimental setup for the GP-based experiments in Figure 1.

| Environment | $T$ [s] | $N$ | $\nu$ [s$^{-1}$] | Algorithm | $\lambda$ | $\beta$ | Learning Rate |
|---|---|---|---|---|---|---|---|
| Pendulum-GP | 2.5 | 12 | 20 | **COMBRL** | 1.0 | – | 0.01 |
| | | | | OCORL | 0 | 7.5 | |
| | | | | PETS | 0 | – | |
| | | | | Mean | 0 | – | |
| MountainCar-GP | 200 | 15 | 1 | **COMBRL** | $10^6$ | – | 0.01 |
| | | | | OCORL | 0 | 30 | |
| | | | | PETS | 0 | – | |
| | | | | Mean | 0 | – | |

Episode horizon $T$, number of episodes $N$, and measurement/control frequency $\nu$ are shared across algorithms for each environment.

For planning, we use the iCEM optimizer (Pinneri et al., 2021) even though it is a discrete-time algorithm. The method iteratively samples action sequences, evaluates them in the dynamics model, and refits a Gaussian distribution to the top-performing ("elite") trajectories. This elite set determines the parameters for the next iteration.

In contrast to vanilla CEM introduced by Rubinstein (1999), iCEM improves efficiency by sampling temporally correlated action sequences and reusing elite trajectories across iterations and MPC steps. We emulate the continuous-time setting by using a fine time discretization (see measurement/control frequency $\nu$ in Table 1) and using the equidistant MSS. The hyperparameters for the planning are given in Table 2.

Table 2: iCEM hyperparameters used for planning in the GP-based experiments.

| Environment | Horizon | # Particles | # Samples | # Elites | Steps | $\alpha$ | Exponent |
|---|---|---|---|---|---|---|---|
| Pendulum-GP | 30 | 10 | 500 | 50 | 10 | 0.2 | 2 |
| MountainCar-GP | 100 | 10 | 500 | 50 | 5 | 0.2 | 2 |

*Horizon* refers to the iCEM planning horizon (in time steps). *Steps* indicates how many CEM optimization iterations are performed per control decision to refine the action distribution using elite samples. The number of time steps and control decisions are given by the measurement/control frequency $\nu$.

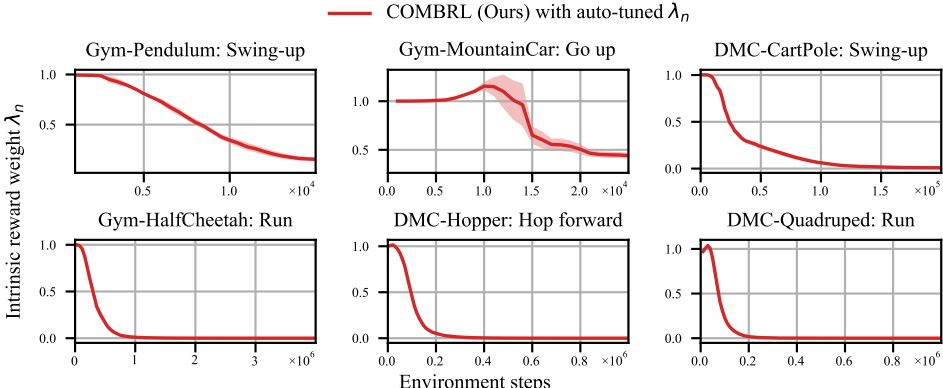

Figure 6: *Evolution of the auto-tuned intrinsic reward weight $\lambda_n$.* We plot $\lambda_n$ as a function of environment steps for COMBRL with the auto-tuning objective in Equation (8) across several Gym/DMC tasks, as seen in Figure 2. Overall, $\lambda_n$ tends to decrease over time, indicating stronger exploration early in training and reduced weighting on the intrinsic reward as learning progresses.

## C.2 Computational costs

We give the computational costs for our GP experiments in Table 3. This shows that the co-optimization over the reward and the optimistic dynamics as well as the reparametrization trick used by the OCORL algorithm are computationally prohibitive, and require around $3\times$ the compute time compared to **COMBRL**.

Table 3: Computation cost comparison (training time) for each algorithm across environments.

| Environment | **COMBRL** | OCORL | Mean | PETS |
|---|---|---|---|---|
| Pendulum-GP [a] | $10.8 \pm 0.13$ min | $30.6 \pm 0.4$ min | $10.4 \pm 0.04$ min | $30.78 \pm 0.27$ min |
| MountainCar-GP [b] | $1.29 \pm 0.01$ h | $4.55 \pm 0.1$ h | $1.28 \pm 0.03$ h | $4.67 \pm 0.16$ h |

[a] Mean total training time. GPU: NVIDIA GeForce RTX 2080 Ti.
[b] Mean training time per episode. GPU: NVIDIA GeForce RTX 2080 Ti.

## C.3 Auto-tuning experiments

In Figure 2, we auto-tune the intrinsic reward weight $\lambda_n$ following the method proposed by Sukhija et al. (2025a), who demonstrate that this approach is effective across a range of model-free off-policy RL methods in discrete time. Specifically, we adjust $\lambda$ by minimizing the loss:

$$L(\lambda) = \underset{\boldsymbol{x} \sim \mathcal{D}_{1:n}, \boldsymbol{u} \sim \boldsymbol{\pi}_n, \bar{\boldsymbol{u}} \sim \bar{\boldsymbol{\pi}}_n}{\mathbb{E}} \log(\lambda)(\boldsymbol{\sigma}_n(\boldsymbol{x}, \boldsymbol{u}) - \boldsymbol{\sigma}_n(\boldsymbol{x}, \bar{\boldsymbol{u}})). \tag{8}$$

Here, $\bar{\boldsymbol{\pi}}_n$ denotes a slowly updated target policy obtained via Polyak averaging of $\boldsymbol{\pi}_n$. The objective promotes larger $\lambda$ values when the current policy explores less than the target policy. Figure 6 shows that the auto-tuned intrinsic reward weight $\lambda_n$ typically decreases over training for several of the tasks shown in Figure 2, indicating that exploration is emphasized early on and gradually reduced as the model and policy stabilize.

## C.4 BNN experiments

In our experiments that do not explicitly use Gaussian Processes, we train an ensemble of 5 neural networks to model forward dynamics. Model epistemic uncertainty is estimated via the disagreement among the ensemble members (Pathak et al., 2019). To further leverage the model, we augment the data by including synthetic transitions. For each policy update, we sample real transitions $(\boldsymbol{x}, \boldsymbol{u}, \dot{\boldsymbol{y}})$ from the replay buffer $\mathcal{D}_{1:n}$ and add corresponding model-predicted transitions $(\boldsymbol{x}, \boldsymbol{u}, \dot{\boldsymbol{y}}')$, where $\dot{\boldsymbol{y}}'$ is generated by the mean model $\boldsymbol{\mu}_n$. This lets us blend real and synthetic rollouts, similar to the strategy used by Janner et al. (2019), thereby increasing the update-to-data (UTD) internal ratio.

The policy is optimized using soft actor-critic (SAC, Haarnoja et al., 2018), an off-policy actor-critic method that learns a stochastic policy by maximizing expected return together with an entropy bonus to encourage exploration and stabilize training. We adopt the same hyperparameters as Sukhija et al. (2025a) for optimizing the loss via stochastic gradient descent in Equation (8) and for configuring the UTD. We also periodically perform soft resets for the policy for training stability (D'Oro et al., 2023). The hyperparameters for the statistical model and SAC are given in Table 4. For the ensemble-based experiments, we use several environments from the Gym and DMC benchmark suites (Brockman et al., 2016; Tunyasuvunakool et al., 2020). We adapt them to the continuous-time setting by approximating the derivatives using a finite difference filter. The measurement/control frequency is given by the duration of an environment step `dt`.

Table 4: Hyperparameters for ensemble-based experiments with SAC in Section 4.

| Environments | Action Repeat | Policy / Critic Architecture | Model Architecture | Learning Rate | Batch Size |
|---|---|---|---|---|---|
| Gym – Pendulum / MountainCar | 1 | (256,256) | 5×(256,256) | $3 \times 10^{-4}$ | 256 |
| Gym – Reacher | 2 | (256,256) | 5×(512,512) | $3 \times 10^{-4}$ | 256 |
| Gym – other environments[a] | 2 | (256,256) | 5×(256,256) | $3 \times 10^{-4}$ | 256 |
| DMC – Quadruped | 2 | (256,256) | 5×(512,512) | $3 \times 10^{-4}$ | 256 |
| DMC – Humanoid | 2 | (512,512) | 5×(512,512) | $3 \times 10^{-4}$ | 256 |
| DMC – other environments[b] | 2 | (256,256) | 5×(256,256) | $3 \times 10^{-4}$ | 256 |

[a] HalfCheetah, Hopper, Pusher.
[b] Hopper, CartPole.

## C.5 DOWNSTREAM TASKS

To evaluate zero-shot generalization in Figures 3 and 4, we introduce custom downstream tasks that differ semantically from the primary training objective. While the primary task corresponds to the default reward in each Gym or DMC environment, the downstream task uses a custom reward function that incentivizes behaviour that contrasts with the original task (e.g., moving away instead of toward a goal).

We implement each downstream task by overriding the reward computation in the Gym or DMC environment using environment wrappers. Table 5 summarizes the evaluated primary and downstream tasks. It also gives the internal reward parameter $\lambda_n$ for the experiments shown in Figure 3. For said experiments, the downstream rewards are defined as follows:

- **MountainCar** – go up left: Encourages the car to reach the leftmost side of the hill, in contrast to the standard goal on the right.
- **HalfCheetah** – run backwards: Reverses the locomotion objective by rewarding backward velocity.
- **Hopper** – hop backwards: Rewards hopping backwards while maintaining a healthy posture.[3]
- **Pendulum**
  - Balance upright: Starts upright and rewards maintaining the upright position.
  - Swing up: Starts with the pendulum pointing downward and rewards swinging it up.
  - Swing down: Starts upright and rewards swinging the pendulum downward.
  - Keep down: Starts downward and rewards staying down.
- **Reacher** – go away: Penalizes proximity to the goal, inverting the standard reaching task.
- **Pusher** – push away from target: Encourages the agent to push the object away from the goal location.

---

[3]The Hopper environment in Gym introduces a `healthy_reward`, which is preserved for the downstream task.

Table 5: Primary and downstream tasks used in our evaluation. Each downstream task is defined via a custom reward that encourages behaviour contrasting the primary task. We also provide the algorithm hyperparameters used in Figure 3, namely the strategy used and the optimized internal reward weight $\lambda_n$.

| Environment | Primary Task | Downstream Task | Strategy[a] | $\lambda$ |
|---|---|---|---|---|
| MountainCar | Go up right | Go up left | Annealing | 50 |
| HalfCheetah | Run forward | Run backward | Annealing | 2 |
| Hopper | Hop forward | Hop backward | Static | 10 |
| Pendulum | Balance upright | Swing down | Annealing | 10 |
| Pendulum | Keep down | Swing up | Annealing | 50 |
| Reacher | Reach target | Keep away | Static | 0.17 |
| Pusher | Push to target | Push away | Static | 0.56 |

[a] Static denotes a fixed internal reward weight $\lambda_n = \lambda$, annealing a decreasing reward weight $\lambda_n \propto \lambda \cdot (1 - n/N)$.

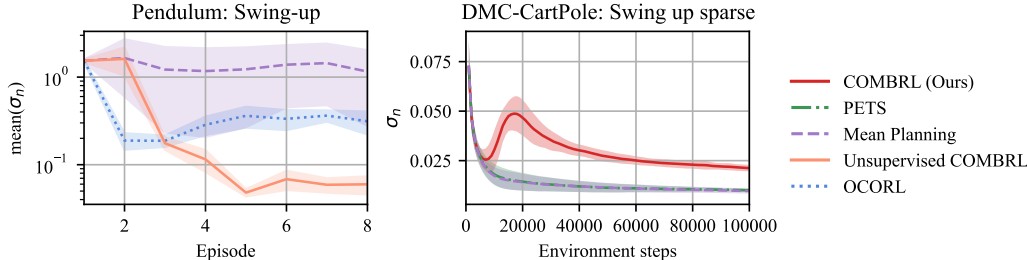

Figure 7: *Epistemic uncertainty under **COMBRL**.* **Left:** *Unsupervised **COMBRL** ($\lambda_n \to \infty$) on Pendulum.* At the end of each episode, we evaluate the model uncertainty $\|\boldsymbol{\sigma}_n(z)\|$ on 100 random samples $z \in \mathcal{X} \times \mathcal{U}$ and track its evolution, showing that unsupervised exploration systematically reduces epistemic uncertainty across the entire reachable state-action space. **Right:** *COMBRL with auto-tuned intrinsic reward weight $\lambda_n$ on DMC-CartPole.* We plot the epistemic uncertainty encountered along the executed trajectories at each environment step. Compared to Mean and PETS, **COMBRL** is guided towards regions of higher epistemic uncertainty while still driving uncertainty down over time, indicating that $\lambda_n$ effectively shapes exploration towards informative state-action regions.

### C.6 EPISTEMIC UNCERTAINTY DECREASE

We visualize how **COMBRL** reduces epistemic uncertainty over time in both the unsupervised and reward-driven regimes in Figure 7.

In the **unsupervised setting**, we run **COMBRL** with a purely intrinsic objective (i.e., $\lambda_n \to \infty$) on the Pendulum environment and monitor epistemic uncertainty over the *entire* reachable state-action space. Concretely, at the end of each episode $n$ we sample 100 *random* state-action pairs $z = (\boldsymbol{x}, \boldsymbol{u})$ from $\mathcal{X} \times \mathcal{U}$ and evaluate the model uncertainty $\|\boldsymbol{\sigma}_n(\boldsymbol{z})\|$ at these points. This shows that unsupervised exploration alone is sufficient to drive down epistemic uncertainty globally over $\mathcal{X} \times \mathcal{U}$, rather than only along the visited trajectories.

In the **reward-driven setting with auto-tuned** $\lambda_n$, we consider the CartPole environment and use the method from Appendix C.3 to auto-tune $\lambda_n$. Here we track the epistemic uncertainty encountered *along the executed trajectories* at each environment step. Compared to the Mean-planning and PETS baselines, **COMBRL** consistently visits regions with higher epistemic uncertainty, indicating that the intrinsic reward term steers the agent towards informative parts of the state-action space, while the overall level of uncertainty still decreases over time.

### C.7 TIME-ADAPTIVE EXPERIMENTS

The time-adaptive TaCoS framework (Treven et al., 2024) generalizes continuous-time RL to settings where sensing and control actions are costly. Instead of interacting with the system at a fixed frequency, the agent actively chooses *when* to sense and apply control inputs, thus adapting the sampling rate over time.

Formally, the dynamics are described by the unknown flow $\boldsymbol{\Phi}^*$, which maps a state-action-time triple to the successor state and the integrated reward:

$$\boldsymbol{\Phi}^*(\boldsymbol{x}, \boldsymbol{u}, t) = (\boldsymbol{x}', b),$$

where $\boldsymbol{x}'$ is the next state at time $t + \Delta t$ and

$$b = \int_t^{t+\Delta t} r(\boldsymbol{x}(s), \boldsymbol{u}(s)) \, ds$$

is the *transition reward*, i.e., the cumulative task reward over the chosen interval $\Delta t$. The length of $\Delta t$ reflects the measurement and control schedule $S$, meaning that fewer but longer intervals correspond to less frequent interaction.

Crucially, in TaCoS each sensing or control action incurs a unit *interaction cost $C$*. The agent must therefore balance two objectives: maximizing cumulative task reward while minimizing the number of costly interactions. This induces a joint optimization problem over both the control policy and the measurement schedule. Transition rewards provide a natural way to capture variable step lengths, allowing the planner to evaluate policies under adaptive sampling schemes.

In the model-based RL setting, we maintain a statistical model $\mathcal{M}_n$ of the flow $\boldsymbol{\Phi}^*$. For the mean and PETS baselines (Mean-TaCoS and PETS-TaCoS), planning is carried out with respect to the task reward only. In contrast, OCORL-TaCoS applies optimism to the dynamics: at episode $n$, the policy $\boldsymbol{\pi}_n$ is selected by maximizing the optimistic value over all plausible models in $\mathcal{M}_{n-1}$.

In this framework, **COMBRL** applies seamlessly. The statistical model $\mathcal{M}_n$ naturally extends to modeling flows with transition rewards, and the intrinsic reward shaping in **COMBRL** integrates epistemic uncertainty with extrinsic task rewards. As a result, **COMBRL**-TaCoS combines optimism-driven exploration with adaptive measurement scheduling, reducing unnecessary interactions while preserving sample efficiency in the adaptive regime.

For the time-adaptive experiments, we evaluate **COMBRL** on two continuous-time environments. We first reuse the **Pendulum-GP** environment described in the GP experiments (see Appendix C.1), with an added interaction cost $C$.

In addition, we include the more challenging **RC Car** environment, using the implementation of Treven et al. (2024). The continuous-time dynamics are integrated with a small base step size (`dt = 1/30s` by default). The reward function combines a tolerance-based state term (encouraging the car to reach a goal pose) with penalties on control effort and smoothness.

The RC Car follows a nonlinear bicycle model with blended kinematic and dynamic components. The state is

$$\boldsymbol{x} = [p_x, p_y, \theta, v_x, v_y, \omega]^\top, \quad \boldsymbol{u} = \begin{bmatrix} \delta \\ d \end{bmatrix},$$

where $(p_x, p_y)$ is the position, $\theta$ the heading angle, $(v_x, v_y)$ the local velocities, and $\omega$ the yaw rate. The control inputs are steering angle $\delta$ and throttle $d$.

The kinematic model is

$$\dot{p}_x = v_x \cos\theta - v_y \sin\theta, \quad \dot{p}_y = v_x \sin\theta + v_y \cos\theta, \quad \dot{\theta} = \omega.$$

The dynamic accelerations are given by

$$\dot{v}_x = \tfrac{1}{m}\Big(f_{r,x} - f_{f,y}\sin\delta + mv_y\omega\Big),$$
$$\dot{v}_y = \tfrac{1}{m}\Big(f_{r,y} + f_{f,y}\cos\delta - mv_x\omega\Big),$$
$$\dot{\omega} = \tfrac{1}{I}\Big(f_{f,y}l_f\cos\delta - f_{r,y}l_r\Big),$$

where $m$ is the car mass, $I$ its moment of inertia, $l_f, l_r$ the distances to the front/rear axles, and $f_{f,y}, f_{r,y}, f_{r,x}$ are the tire forces determined by the nonlinear Pacejka tire model.

The blended model used in practice interpolates between the kinematic and dynamic models depending on the velocity regime (see the implementation by Treven et al. 2024 for details).

We give the relevant parameters for the time-adaptive experiments in Table 6. In Figure 5, we show the final performance at convergence of the evaluated algorithms. For completeness, we also offer the complete learning curves in Figure 8.

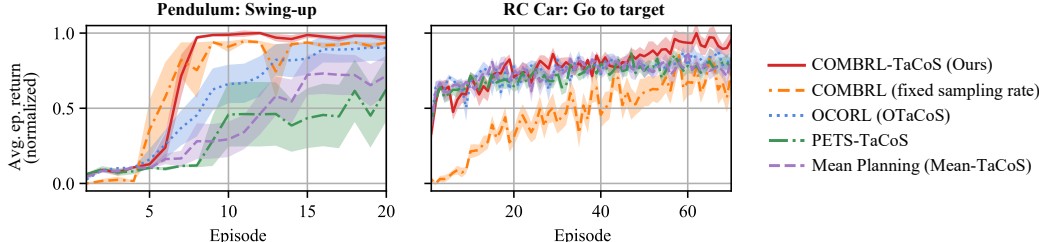

Figure 8: Learning curves of **COMBRL**-TaCoS compared to OTaCoS, Mean-TaCoS, PETS-TaCoS, and COMBRL with fixed control rate (equidistant MSS). Results are averaged over 10 random seeds, with mean and standard error shown. **COMBRL**-TaCoS achieves competitive or superior returns while requiring fewer interactions than its fixed-rate variant, and matches or exceeds the performance of OTaCoS.

Table 6: Experimental setup with environment details and hyperparameters for the TaCoS experiments.

|  | Pendulum | RC Car |
| --- | --- | --- |
| Episode horizon $T$ [s] | 10 | 3 |
| Number of episodes $N$ | 20 | 70 |
| Base step size $\Delta t$ [s] | 0.05 | 1/30 |
| Max steps per episode | 200 | 100 |
| Min steps per episode | 40 | 20 |
| Interaction cost $C$ | 0.1 | 0.4 |
| Confidence level $\beta$ | 2 | 2 |
| Internal reward weight[a] $\lambda$ | 1 | 10 |
| Model architecture | $5\times(64,64,64)$ | $5\times(64,64,64)$ |
| Policy hidden layers | $(64,64)$ | $(64,64)$ |
| Learning rate | $3 \times 10^{-4}$ | $3 \times 10^{-4}$ |
| Batch size | 256 | 256 |

[a] We use an annealing reward weight schedule, i.e., $\lambda_n \propto \lambda \cdot (1 - n/N)$.

