# OpenReview forum: "Sample-efficient and Scalable Exploration in Continuous-Time RL"
_ICLR.cc/2026/Conference — ICLR 2026 Poster_

### Official Review · Reviewer_LpWF · 2025-10-21

**Soundness:** 3
**Presentation:** 2
**Contribution:** 3
**Rating:** 4
**Confidence:** 3

**Summary:**

The paper introduces COMBRL, a continuous-time optimistic model-based RL framework that learns probabilistic ODE dynamics and plans policies by maximizing a reward-plus-uncertainty objective governed by a single $\lambda$ schedule. They analyze both reward-driven and unsupervised (intrinsic-only) regimes, proving sublinear regret and sample-complexity bounds, and demonstrate on Gymnasium and DeepMind Control benchmarks that COMBRL improves sample efficiency over OCORL, PETS, and mean-planning baselines while also handling time-adaptive sensing and control.

**Strengths:**

- Clean optimism formulation: a single $\lambda$ schedule unifies reward-driven and intrinsic exploration, allowing practitioners to traverse between task-focused and unsupervised regimes.
- Theory for both regimes: sublinear regret for the reward-driven case and sample-complexity guarantees for pure intrinsic learning extend discrete-time optimism analysis to continuous-time ODE settings.
- Model-agnostic dynamics learning: the framework accommodates Gaussian processes, Bayesian neural nets, ensembles, and other uncertainty-aware estimators, and remains solver/discretization agnostic through its continuous-time formulation.
- Empirical coverage: experiments span standard Gymnasium and DMC control tasks, include time-adaptive sensing/control, and report downstream-task performance gains using the learned models, showing consistent sample-efficiency improvements over OCORL, PETS, and mean-planning baselines.

**Weaknesses:**

- Limited scope of benchmarks: results stop at modest Gymnasium/DMC tasks; harder domains (e.g., Humanoid, Dog) and sparse-reward environments (e.g., Maze2D) that would stress exploration are absent, so the empirical evidence for better exploration remains less convincing.
- Missing modern baselines: comparisons exclude recent model-free optimistic or exploration-focused controllers such as BRO [1], so it is unclear whether COMBRL advances the empirical state of the art beyond older baselines like OCORL and PETS.
- $\lambda$ sensitivity and exploration role not sufficiently explored: tables in the Appendix show $\lambda$ varying considerably across tasks. Plots seem to show contradictory results for how exploration strength impacts performance or when higher bonuses help versus hurt.

[1] Nauman et al., Bigger, Regularized, Optimistic: scaling for compute and sample-efficient continuous control, 2024.

**Questions:**

- In the fixed-rate experiments, do the authors use the default Gym/DMC sampling interval, or do they employ a finer measurement-selection schedule? Please quantify any difference (e.g., actual step size) and discuss how sensitive COMBRL is to noisy or finite-difference estimates of $\dot{x}$ under that schedule.
- Table 1 and Table 5 list quite different $\lambda$ values. Figure 2 credits intrinsic bonuses for COMBRL’s gains, yet Figure 4’s ablation sometimes peaks at $\lambda = 0$, suggesting exploration may hurt. Could the authors reconcile these results by clarifying how the hand-tuned and auto-tuned schedules were obtained, explaining when exploration helps versus harms (e.g., Hopper), and showing how the auto-tuned $\lambda$ evolves over time (plots)?
- COMBRL’s continuous-time focus is compelling, yet BRO [1] remains the state-of-the-art optimistic model-free continuous-control method with a similar exploration rationale—what prevents a direct comparison to show whether COMBRL’s advantages persist against that baseline?
- Since the method hinges on epistemic bonuses, could the authors provide plots showing how model uncertainty decreases over episodes, especially in the unsupervised regime?
- Given the exploration focus, could the evaluation be extended to harder DMC benchmarks such as Humanoid-run or Dog-run, and to sparse-reward settings like Maze2D, to substantiate the claimed advantages beyond the current tasks?

I find the continuous-time and unsupervised RL framing compelling; if the authors can address the questions above, I would be inclined to raise my score.

[1] Nauman et al., Bigger, Regularized, Optimistic: scaling for compute and sample-efficient continuous control, 2024.

---

> ### Author Response · Authors · 2025-11-21
>
> Thank you very much for taking the time for the detailed review of our work!
>
> **Harder domains and real-world relevance.**
> We are actively working on applying COMBRL to harder domains and real-world systems, and see this as a central direction for future work. Compared to previous theoretically backed continuous-time methods, COMBRL is significantly more scalable and practical. In the revision, we now include additional experiments on the DMC Humanoid [1], which empirically support that COMBRL scales to higher-dimensional, robotics-style problems.
>
> **BRO and baseline choice.**
> BRO [2] is an off-policy *discrete-time* RL algorithm. Since our focus is **continuous-time** RL, we primarily compare to OCORL and TaCoS, which, to the best of our knowledge, are the state-of-the-art regarding both theoretical guarantees and practical implementations. Moreover, [4] already show that continuous-time RL methods can outperform discrete-time approaches even when the true dynamics are known, which further motivates focusing on continuous-time baselines. That said, we agree that BRO’s BRONET architecture is applicable: It can be used as the statistical model for our unknown dynamics $\boldsymbol{f}^\*$. We are willing to test this architecture and will do our best to provide our findings during this paper discussion phase.
>
> **$\lambda$ sensitivity and exploration.**
> We agree that $\lambda$ plays a crucial role. For large $\lambda$, COMBRL recovers the unsupervised exploration regime for which we provide guarantees in Theorem 2. For the **single-task** setting (Fig. 2), we use the auto-tuning scheduler [3], which adapts $\lambda$ similarly to the entropy coefficient in SAC; we find this works well in our continuous-time setting and leads to efficient exploration in sparse-reward and high-dimensional tasks.
>
> In contrast, Fig. 3-4 focus on a **multitask** setting (primary + secondary downstream task). Here, too much intrinsic reward can hurt performance on the primary task while still improving the secondary (unseen) task. To study this trade-off, we therefore **hand-tune** $\lambda$ per environment via a small grid search and explicitly ablate its effect: Fig. 4 shows that modest $\lambda$ can yield substantial gains on the secondary task while maintaining good performance on the primary one. In our view, this highlights that intermediate $\lambda$ values effectively balance goal-directed behaviour with model-uncertainty reduction. Because the auto-tuning scheduler from [3] is designed for a single task, its automatic tuning does not directly extend to our multitask case, and we regard principled multitask $\lambda$-scheduling as an interesting direction for future work.
>
> The wide range of $\lambda$ values across tasks is largely due to differing scales between extrinsic rewards and intrinsic bonuses both within and across environments. An alternative parametrization such as $(1-\lambda)\cdot r_{\text{ext}} + \lambda\cdot r_{\text{int}}$ with $\lambda \in [0,1]$ would only reparameterize this trade-off but not fundamentally change the nature of the weighted sum.
>
> **Fixed-rate experiments (MSS and step size).**
> In all fixed-rate experiments we use the **default Gym/DMC integration step** as the sampling interval for the equidistant measurement-selection strategy (MSS), i.e., one measurement per environment step without finer sub-stepping. The **control** interval is determined by the `action_repeat` parameter reported in Table 4: for `action_repeat = k`, we query the policy every k-th measurement and apply zero-order hold to keep the same action for the k intermediate simulator steps. For the Gym/DMC experiments, we use finite-difference estimates between simulator steps to approximate $\dot{x}$. We will clarify this in the appendix.
>
> **Logging $\lambda$ and $\sigma_n$ over time.**
> We agree that visualizing $\lambda_n$ and the evolution of epistemic uncertainty is valuable. In the revised appendix we will add plots showing the decrease of $\lambda$ and $\sigma_n$ over episodes (in both reward-driven and unsupervised regimes). For now, in the Appendix we provide a new Figure 6 in Appendix C which shows the decrease of $\sigma_n$ for two experiments.
>
> We hope this addresses your concerns. If so, we would greatly appreciate an updated score; otherwise, we are happy to further clarify any remaining questions.
>
> **References:**
> [1] Google DeepMind Control Suite, 2018.
> [2] Bigger, Regularized, Optimistic: scaling for compute and sample-efficient continuous control, NeurIPS, 2024.
> [3] MaxInfoRL: Boosting exploration in reinforcement learning through information gain maximization, ICLR, 2025.
> [4] Efficient exploration in continuous-time model-based reinforcement learning, NeurIPS, 2023.

---

> > ### Comment · Reviewer_LpWF · 2025-11-25
> >
> > Thank you for the detailed rebuttal and additional experiments; several clarifications were helpful. I am still not fully convinced of the following:
> > - Thanks for the additional Humanoid experiment, but the rationale for omitting BRO remains unconvincing. The rebuttal cites [4] to argue that continuous-time methods can outperform discrete ones, yet the newly reported Humanoid scores remain well below those achieved by BRO, so there is a contradiction between [4] and your results. Also, to the best of my knowledge PETS is a discrete-time method, so the reasoning for omitting BRO remains unconvincing.
> > - The exploration-focused evaluation remains limited to shaped-reward Gym/DMC tasks. I think the work would benefit substantially from explicitly evaluating on sparse-reward settings (e.g., Maze-style environments with “goal or zero” feedback) to show that COMBRL’s exploration advantages carry over when the learning signal is genuinely weak.

---

> > > ### Author Response · Authors · 2025-12-03
> > >
> > > Thank you for the follow-up and for pointing this out. Our empirical comparisons focus on continuous-time baselines with theoretical backing (OCORL and TaCoS, which are the state-of-the art for continuous-time RL). We additionally use model-based planners (Mean / PETS). For PETS, we modify the trajectory sampling scheme so that rollouts are drawn from our model of $\boldsymbol{f}^\*$, just as the Mean planner plans with the mean $\boldsymbol{\mu}_n$. Thus, the PETS variant we use is effectively a continuous-time version.
> > >
> > > In a similar fashion, BRO and BRONET are designed for discrete-time control and would need to be adapted to a continuous-time analogue to ensure a fair comparison. We ran preliminary experiments where we naively replaced our critic with a BRONET-style network but did not observe consistent gains, which we suspect is due to this mismatch between discrete-time assumptions and our continuous-time ODE formulation.
> > >
> > > We believe that the use of continuous-time RL mainly is motivated by real-world control systems, from physical robots to biological processes, which are naturally modeled by continuous-time dynamics governed by ordinary differential equations (ODEs). ATARI-style maze environments, on the other hand, are inherently discrete time, so they are somewhat orthogonal to our continuous-time focus and we chose to prioritize ODE-based control benchmarks in this work. Furthermore, in Figure 2, we in fact explicitly evaluate COMBRL on sparse-reward tasks.
> > >
> > > Overall, we believe the additional experiments and clarifications in our author responses reinforce our core claims about COMBRL’s theoretical guarantees, scalable optimism mechanism, and its ability to trade off reward-driven and intrinsic exploration in continuous-time RL.

---

### Official Review · Reviewer_8afj · 2025-10-27

**Soundness:** 3
**Presentation:** 3
**Contribution:** 3
**Rating:** 6
**Confidence:** 2

**Summary:**

This paper studies the continuous-time model-based RL problem where the system dynamics are represented by nonlinear ODEs. The proposed algorithm, COMBRL, is built on uncertainty-aware probabilistic models (GPs or BNNs) and aims to optimize a reward-plus-uncertainty objective in continuous time. The work focuses on two settings: reward-driven RL and unsupervised RL. In experiments, the proposed COMBRL achieves strong empirical results across different continuous-time RL benchmarks and is sample-efficient compared to other baselines.

**Strengths:**

- I am not an expert in this area but I agree that the study of the continuous-time RL is promising. In addition, this work further explores the unsupervised setting, which is important and interesting.
- The reward-plus-uncertainty objective makes sense and the results look good.

**Weaknesses:**

1. My main concern is that the continuous-time RL is already studied in previous work such as [1], in which the system is also modelled by ODEs and the model is built on GP. This makes the novelty incremental.
2. In Figure 2, the baseline approaches are only mean planner and PETS. It would be better to provide more comparisons with other state-of-the-art methods on these GYM and DMC tasks.




[1] Efficient exploration in continuous-time model-based reinforcement learning, NeurIPS 2023.

**Questions:**

The current settings and tasks are too simple, I wonder can this algorithm be applied to some real-world problems?

---

> ### Author Response · Authors · 2025-11-21
>
> Thank you very much for taking the time to review our work! Regarding your two comments, we would like to answer in the following:
> 1. Compared to [1], both COMBRL and the method proposed in [1] provide theoretical guarantees of sublinear regret and rely on the principle of optimism in the face of uncertainty. However, the algorithm in [1] is computationally impractical in its exact form and must be approximated using the reparametrization trick together with additional control variables. This approach only works well for low-dimensional state spaces and is computationally expensive, as shown in Figure 1 and Appendix C.2. In contrast, COMBRL takes a more practical approach to exploration by combining extrinsic with intrinsic rewards for exploration. This approach is directly implementable, scales to high-dimensional systems, and introduces minimal computational overhead, while enjoying the same theoretical guarantees as [1].
>
> 2. We compare our method against OCORL and TACOS (Figure 5), which, to the best of our knowledge, are state-of-the-art (SOTA) baselines for continuous-time RL in terms of both theory and practice. Moreover, the crucial contribution of our work is to show that we can avoid the intractable optimization from OCORL and TACOS induced from optimism by simply combining the intrinsic rewards with extrinsic ones. For most of the environments considered in our work, OCORL is intractable, and we outperform it in the low-dimensional tasks (Figure 1). Therefore, we focus on isolating the benefits of intrinsic exploration in our remaining experiments. To this end, we compare with mean and pets planning for learning, which are scalable yet greedy exploration techniques for extracting a policy from a learned model.
>
> Regarding your **question**:
> We are actively working on applying our method in the real world. We see this as a promising direction for future work. Compared to previous theoretically backed methods, COMBRL is much more scalable and practical. From our experiments with the algorithm, we believe that it can also work in higher-dimensional and real-world problems. To further show scalability, We now provide two additional experiments on the Humanoid robot from the DMC suite [1]. We believe this further shows that COMBRL scales to higher-dimensional problems.
>
> We hope we have fully addressed your concerns. If so, we would greatly appreciate an updated score; otherwise, please let us know if there is anything further we can clarify.
>
> __References:__
> [1] Treven et. al., Efficient exploration in continuous-time model-based reinforcement learning. NeurIPS, 2023.

---

> > ### Comment · Reviewer_8afj · 2025-11-26
> >
> > I appreciate the authors' efforts in the rebuttal. Most of my concerns are addressed and thus I keep my original positive score.

---

### Official Review · Reviewer_F13n · 2025-10-31

**Soundness:** 3
**Presentation:** 3
**Contribution:** 3
**Rating:** 4
**Confidence:** 3

**Summary:**

This paper introduces COMBRL, a continuous-time optimistic model-based reinforcement learning algorithm designed to balance task performance and model exploration. The authors establish theoretical performance guarantees and validate the approach empirically across a suite of continuous-time deep RL benchmarks.

**Strengths:**

The theoretical studies are solid.

**Weaknesses:**

The primary weakness is the numerical evaluation. When ground-truth values are available, experiments should report coverage rates of the true value; limiting results to the averaged returns and standard deviations does not provide an adequate assessment.

**Questions:**

See the weakness. The author may consider to work on a simulated environment for which the true values are know, and experiments should report covedrage rates of the true value.

---

> ### Author Response · Authors · 2025-11-21
>
> Thank you very much for your review. Regarding the weakness in the numerical evaluation:
>
> For all dynamical systems considered in the paper, we do have access to simulators. However, due to the nonlinear nature of the dynamics, the optimal policies (ground truth) are not available in closed form for any of these systems. In our GP experiments, we now additionally report the best performance achieved by MPC (iCEM) on the true system. This baseline assumes that the dynamical system $\boldsymbol{f}^\*$ is known and plans with respect to the task/reward at hand subject to the true dynamics. This acts as an estimate for the performance of the optimal policy $\boldsymbol{\pi}^\*$.
>
> We hope we have fully addressed your concerns. If so, we would greatly appreciate an updated score; otherwise, please let us know if there is anything further we can clarify.

---

### Official Review · Reviewer_G4x3 · 2025-11-01

**Soundness:** 3
**Presentation:** 2
**Contribution:** 2
**Rating:** 4
**Confidence:** 2

**Summary:**

The paper addresses continuous-time reinforcement learning (RL), where system dynamics are modeled by unknown nonlinear ordinary differential equations (ODEs). The authors propose COMBRL, a model-based RL algorithm that uses probabilistic models to learn an uncertainty-aware approximation of the ODE dynamics. COMBRL balances maximizing extrinsic reward and reducing model epistemic uncertainty, enabling efficient exploration.

**Strengths:**

- Continuous-time model-based RL is an under-explored area with practical relevance, e.g., in robotic control.
- The paper includes extensive experiments evaluating design choices and hyperparameter sensitivity across multiple benchmarks.
- Theoretical guarantees are provided.

**Weaknesses:**

- In Figure 3, COMBRL’s performance on the primary task is modest: in 5 out of 7 tasks, its error bars overlap substantially with those of the Mean Planning baseline, raising questions about practical gains.
- The paper assumes familiarity with several methods (e.g., SAC, iCEM, and TaCoS) without brief introductions, which hinders accessibility.
- The novelty is unclear. While continuous-time RL is valuable, the core idea—-balancing reward maximization and epistemic uncertainty in the objective function—-is not novel. The contribution appears to be an integration of existing components rather than a conceptual advance. Clarifying the key insight beyond "applying model-based RL to continuous time" would strengthen the paper.
- Algorithm 1 lacks sufficient detail. Critical aspects, such as how the optimistic objective is solved, how rollouts are collected, and how the dynamics model is updated, are omitted. Including a more descriptive outline would improve reproducibility.

**Questions:**

- How were hyperparameters (e.g., the trade-off coefficient $\lambda$) set (static or annealing) for the experiments in Figure 3? Was tuning performed per environment, and if so, what protocol was used?

---

> ### Author Response · Authors · 2025-11-21
>
> Thank you very much for your review! Regarding your comments:
>
> - **Figure 3:** While COMBRL comes with theoretical guarantees for converging to an optimal controller, we are not aware of any prior work that provides such guarantees for Mean planning in the nonlinear setting. In practice, many environments have sufficiently well-shaped rewards for Mean planning to perform reasonably well. However, in sparse-reward scenarios, Mean planning typically fails to discover good policies (see, for instance, MountainCar “Go Up” or CartPole “Swing-up” in Figure 2).
> Since COMBRL offers theoretical guarantees of optimality with only minimal computational overhead relative to Mean planning, we believe this results in meaningful practical benefits that practitioners will appreciate.
>
> - **Acronyms explanation:** Thank you for pointing this out. We will add a description of the planning methods used (SAC, iCEM, TACOS) in the appendix.
>
> - **Unclear novelty:** We agree that the high-level idea of balancing extrinsic and intrinsic rewards has been studied by prior work in the context of discrete-time RL. However, to the best of our knowledge, we are the first to propose, study, and provide theoretical guarantees for it in the continuous-time case. Compared to the discrete-time setting, in the continuous-time case, careful consideration of when to control and observe the system is required for the theoretical guarantees. Crucially, while in discrete-time RL the frequency of controlling and observing the system is prespecified by the problem designer, for continuous time, this is chosen by the underlying RL algorithm. This flexibility in the algorithm yields even higher performance than discrete-time methods, as shown in Figure 5. Hence, our theoretical guarantees require additional consideration of the measurement selection/control strategy of the algorithm when theoretically studying the regret (see Section 3.3). In addition, we also give guarantees for the pure exploration method (Theorem 2), which, to the best of our knowledge, we are the first to provide for continuous-time RL.
> Finally, we present a comprehensive empirical study examining how different balancing factors (\lambda) influence performance on both the current task and downstream tasks, i.e., extrinsic vs intrinsic exploration (Figures 3 and 4).
>
> - **Implementation details:** Thank you for highlighting this. We will add further implementation details to the appendix.
>
> Regarding your question on  **$\lambda$ in Figure 3:**
> The scalar trade-off coefficient $\lambda$ was tuned separately for each environment using a manual grid search over a logarithmic grid (e.g. $\lambda\in\{0.1, 1, 10, 100\}$). For each candidate $\lambda$, we trained COMBRL with several random seeds under a fixed interaction budget and selected the $\lambda$ that maximized the final primary-task return. The selected $\lambda$ values (and schedule type where applicable) are reported in Table 4.
>
> We hope we have addressed your concerns. If so, we would greatly appreciate an updated score; otherwise, please let us know if there is anything further we can clarify.

---

### Author Response · Authors · 2025-11-21
**First official comment and paper update**

We thank all reviewers for their comments. We provide an updated version of our paper with additional experiments:
- One reviewer requested how COMBRL performs compared to the ground truth, i.e., the optimal policy. In the GP experiments (Figure 1), we now additionally provide the best performance achieved by MPC (iCEM) on the true system as the reference. This baseline assumes that the dynamical system $\boldsymbol{f}^\*$ is known and plans with respect to the task/reward at hand subject to the true dynamics. This acts as an estimate for the performance of the optimal policy $\boldsymbol{\pi}^\*$, which due to the nonlinear nature of the considered dynamical systems is not available in closed form.
- Two reviewers requested for COMBRL to be evaluated on harder domains. We now provide two additional experiments on the Humanoid robot from the DMC suite [1]. We believe this further shows that COMBRL scales to higher-dimensional problems.
- For transparency, we would like to make the reviewers aware of an oversight on our behalf. For the DMC-Quadruped run experiment in Figure 2, the labels for COMBRL, Mean and PETS were rotated. We noticed this error when running the higher-dimensional experiments mentioned above. This is corrected in the new version. We apologize for any confusion or inconvenience caused. Note that all other experiments in Figure 2 aside from the DMC-Quadruped are unaffected.
- On the request of one reviewer, we added a new Figure 6 in Appendix C.6 that visualizes how COMBRL reduces epistemic uncertainty over time.
- Following the further feedback of the reviewers, we are planning on conducting additional experiments using the BROnet architecture [2], as well as additional plots showing the decrease in the scalar $\lambda$ when using the auto-tuning approach from [3]. We will provide our findings as soon as possible during this paper discussion phase. Thank you for your understanding and patience!

Please let us know if there is anything else we can clarify. Best wishes, the authors.

__References:__
[1] Google DeepMind Control Suite, 2018.
[2] Nauman et. al., [Bigger, Regularized, Optimistic: scaling for compute and sample-efficient continuous control](https://arxiv.org/abs/2405.16158). NeurIPS, 2024.
[3] Sukhija et. al., [MaxInfoRL - Boosting exploration in reinforcement learning through information gain maximization](https://proceedings.iclr.cc/paper_files/paper/2025/file/bd996108ed57d388866ca6deb7acf6cb-Paper-Conference.pdf). ICLR, 2025.

---

### Author Response · Authors · 2025-12-03
**Official comment by authors and second paper update**

We would like to briefly summarize the main updates and clarifications for the AC. We have uploaded a revised version of the paper reflecting these changes.

**New baseline and harder domains**
- We provide additional experiments (see previous author comment for details) comparing COMBRL to an estimate of the optimal policy’s performance in Figure 1, by adding an MPC baseline (iCEM) that plans with access to the true dynamics. This shows that COMBRL attains a substantial fraction of the optimal performance while only using learned models.
- To address concerns about scalability, we also added experiments on the DMC Humanoid robot, showing that COMBRL remains stable, sample-efficient, and scales to higher-dimensional continuous-time control problems.
- Together with the strong performance of the unsupervised variant of COMBRL on previously unseen downstream tasks (Figures 3/4) and the improved sample-efficiency compared to discrete sampling and control schemes (Figure 5), we believe the updated experiments more clearly support the main claims of the paper. In particular, *(i)* COMBRL provides a practical and scalable realization of optimism in continuous-time RL, *(ii)* the single scalar $\lambda_n$ trades off reward-driven and intrinsic exploration with accompanying theoretical guarantees, and *(iii)* the resulting models transfer effectively to harder tasks and flexible sampling strategies.

**Epistemic uncertainty**
In response to questions about uncertainty reduction, we added a new Fig. 6 (Appendix C.6) that visualizes epistemic uncertainty under COMBRL in two cases:
1. an unsupervised Pendulum setting, where uncertainty over the reachable state-action space decreases globally over episodes, and
2. a CartPole setting with auto-tuned $\lambda_n$, where the intrinsic reward steers the agent into uncertain regions while overall uncertainty still decreases along the trajectories.

**Positioning and baselines**
In our rebuttal comments, we clarify the conceptual novelty beyond “adding intrinsic rewards”: COMBRL provides *(i)* a flexible method that trades off reward-driven and intrinsic exploration in the reward-driven setting as well as principled exploration in the unsupervised setting for continuous-time RL, *(ii)* regret and sample-complexity guarantees that explicitly depend on the measurement selection strategy, which, to the best of our knowledge, is the first such result for an intrinsic-reward-based method in continuous-time RL, and *(iii)* overall, a scalable optimism mechanism that avoids the intractable dynamics co-optimization used in previous state-of-the-art methods while retaining similar guarantees.

Our empirical comparisons focus on continuous-time baselines with theoretical backing (OCORL, TaCoS) and scalable model-based planners (Mean / PETS). We also ran preliminary experiments where we replaced our critic with a BRONET-style network but did not observe consistent gains, which we suspect is due to the mismatch between BRO’s discrete-time design and our continuous-time ODE formulation. From our qualitative experience, a principled continuous-time adaptation is non-trivial.

We hope this summary helps contextualize our changes and how they address the main points raised in the reviews. We are, of course, happy to clarify any remaining questions.

---

### Meta-Review · Area_Chair_m1A1 · 2026-01-07

**Summary:**

This paper studies continuous-time reinforcement learning, where the environment dynamics are governed by unknown non-linear ODEs. The main contribution is an algorithm, called COMBRL, which aims at maximizing a weighted sum of extrinsic rewards and model uncertainty. The learning efficiency of COMBRL is established by deriving a sublinear regret bound (in the reward-driven setting) and a sample complexity bound (in no-extrinsic-reward setting). Its empirical performance and scalability are demonstrated by numerical experiments conducted on some RL tasks that are deemed relevant in the context of continuous-time RL.

The reviewers confirm that continuous-time RL is an important, yet under-explored, area with practical relevance, and that the paper makes solid and mathematically rigorous contributions to the field by introducing a provably efficient algorithm --COMBRL-- with performance guarantees in settings with extrinsic. Further, that COMBRL supports the unsupervised setting was appreciated. The reviewers raised comments regarding potentially missing baselines in the empirical evaluations, clarification of technical novelty, missing implementation details as well as some presentational comments.

After carefully reading the reviews and the rebuttal, I conclude that most key concerns were precisely and adequately addressed, and it was further supported by complementary empirical results in the revision. In view of technical soundness of the work and its potential to offer a viable method with practical relevance, as demonstrated through empirical evaluations, I recommend acceptance.

**Reviewer Concerns:**

__Clarification of novelty.__ It was raised that the core idea behind COMBRL is not novel, although it hinges on a sound and viable construction. Although the general idea behind the algorithm is not fundamentally novel, its execution in the context of continuous-time RL under nonlinear ODEs involve highly non-trivial steps due to the need for additional consideration of when to observe and control the system. Such considerations pose challenges in the theoretical performance analysis. In summary, the novelty is further clarified and well justified.

__Missing baselines and more complex domains.__ The reviewers raised comments regarding missing baselines and used environments in the experiments. The revision includes adds experiments using new baselines notably including iCEM, which employs MPC, and BRO. The latter is arguably less relevant, since it is originally a discrete-time RL method. Further, the revised includes experiments on DMC Humanoid robot to showcase the scalability of COMBRL. Overall, I conclude that this issue is settled.

__Presentational issues.__ Such comments are addressed well in the rebuttal.

__Missing details regarding implementations (optimization and dynamics update).__ This includes questions regarding selection/tuning of $\lambda$, its sensitivity, evolution of model uncertainty, etc. These were addressed well in the rebuttal and the revision.

**Reviewer Scores:**

- Reviewer G4x3: The concerns raised by the reviewer were sufficiently and precisely addressed in the rebuttal, in my view. Thus, I expect the reviewer would likely increase their score (to 6).
- Reviewer F13n: The provided review is short, and the only raised concern was addressed well in the rebuttal and revised paper. I therefore have to step in and thus predict that the score would likely be increased to 6 --otherwise I would have to put a low weight on the original rating.
- Reviewer 8afj: The reviewer is already positive. They said that the score will be maintained.
- Reviewer LpWF: The reviewer provided a thorough review and raised key concerns. The rebuttal did a good job on addressing them. However, the reviewer still thinks that BRO, as a baseline, must be included. I think the reviewer must have become more positive after the current state of the rebuttal. Further, I agree with the authors that BRO renders less relevant than the readily included baselines from the literature. Therefore, I conclude that the score must have been increased.

---

### Decision · Program_Chairs · 2026-01-26

Accept (Poster)